# Generation of an expandable intermediate mesoderm restricted progenitor cell line from human pluripotent stem cells

Nathan Kumar[1], Jenna Richter[1], Josh Cutts[2], Kevin T Bush[3], Cleber Trujillo[3], Sanjay K Nigam[1,3], Terry Gaasterland[4], David Brafman[2]*, Karl Willert[1]*

[1]Department of Cellular and Molecular Medicine, University of California, San Diego, San Diego, United States; [2]School of Biological and Health Systems Engineering, Arizona State University, Tempe, United States; [3]Department of Pediatrics, University of California, San Diego, San Diego, United States; [4]Scripps Institution of Oceanography, Scripps Genome Center, University of California, San Diego, San Diego, United States

**Abstract** The field of tissue engineering entered a new era with the development of human pluripotent stem cells (hPSCs), which are capable of unlimited expansion whilst retaining the potential to differentiate into all mature cell populations. However, these cells harbor significant risks, including tumor formation upon transplantation. One way to mitigate this risk is to develop expandable progenitor cell populations with restricted differentiation potential. Here, we used a cellular microarray technology to identify a defined and optimized culture condition that supports the derivation and propagation of a cell population with mesodermal properties. This cell population, referred to as intermediate mesodermal progenitor (IMP) cells, is capable of unlimited expansion, lacks tumor formation potential, and, upon appropriate stimulation, readily acquires properties of a sub-population of kidney cells. Interestingly, IMP cells fail to differentiate into other mesodermally-derived tissues, including blood and heart, suggesting that these cells are restricted to an intermediate mesodermal fate.

*For correspondence:
david.brafman@asu.edu (DB);
kwillert@ucsd.edu (KW)

**Competing interests:** The authors declare that no competing interests exist.

## Introduction

Human pluripotent stem cells (hPSCs; including human embryonic stem [hES] cells and human induced pluripotent stem [hiPS] cells) have the potential to generate the various cell types of the adult body. With their capacity to expand indefinitely, hPSCs provide a potentially unlimited source of mature cell types that can be used for disease modeling, drug discovery, and regenerative medicine purposes. Current methods for generating these therapeutically relevant cell types follow a linear approach in which hPSCs are differentiated in small, discrete steps that mimic the sequence of events occurring during development. The initial step typically involves specification of hPSCs into one of the three embryonic germ layers—ectoderm (EC), endoderm (EN), or mesoderm.

In the case of mesoderm, several protocols have been developed to derive mature tissues stemming from this lineage, including muscle, blood, and urogenital cells (*Kee and Reijo Pera, 2008*; *Ng et al., 2008*; *Lian et al., 2012*; *Taguchi et al., 2014*). While these studies have demonstrated the potential of hPSC-derived mesodermal tissues for cell replacement therapies, these protocols result in the generation of heterogeneous cell populations, some with tumor forming potential, which limits their clinical utility. Additionally, because of the inefficiency of these established protocols, large numbers of input cells are necessary to generate cell types in the quantities necessary for clinical applications.

**eLife digest** The development of 'human pluripotent stem cells' has the potential to revolutionize the future of medicine. This is because these cells can both replicate themselves indefinitely (i.e., they can self-renew) and develop into any of the cell types found in the human body (a process that is referred to as differentiation). These abilities mean that the cells could in theory be used to replace any tissues or organs that have been damaged by disease or injury. Unfortunately, transplanting stem cells that are capable of developing into any type of cell comes with the significant risk that these cells will form into a tumor.

Once a cell has started to differentiate it can typically only go on to generate a restricted number of cell types. However, these differentiating cells also generally lose their ability to self-renew. Kumar et al. set out to challenge this fundamental property of differentiating cells. A high throughput-screening approach was used to test thousands of combinations of bioactive molecules (i.e., molecules that are known to affect living cells in different ways) to identify some that could promote the self-renewal of cells with a restricted potential to differentiate.

Kumar et al. found specific conditions that could cause a population of cells, which they referred to as 'intermediate mesodermal progenitor cells' (or IMP cells for short), to self renew. These cells resemble those found in the middle layer of a very early human embryo, which typically go on to develop into only a subset of tissue types in the body—for example, muscle, kidneys and blood vessels, but not brain or lungs. Yet, when Kumar et al. stimulated the self-renewing IMP cells, these cells only differentiated into the cell types that make up the kidney and not any other types of cell.

This tight restriction on the differentiation potential of these cells is highly important, because it means that these cells could greatly advance methods to generate kidney cells or even whole kidneys in the laboratory that are suitable for transplantation.

Expansion of intermediate progenitor populations of differentiating hPSCs followed by subsequent differentiation is an alternative approach for generating highly enriched and well-defined cell populations required for cell-based therapies and disease modeling. For example, homogenous, expandable ectodermally- and endodermally-restricted progenitor populations have been generated from hPSCs (*Reubinoff et al., 2001*; *Shin et al., 2006*; *Chambers et al., 2009*; *Cheng et al., 2012*). However, similar methods to generate cell types restricted to the mesodermal lineage have yet to be developed.

The cell microenvironment plays a critical role for regulating self-renewal and differentiation of many progenitor cell populations that exist within the developing and fully mature adult organism (*Moore and Lemischka, 2006*; *Jones and Wagers, 2008*). In this study, we used a multifactorial high-throughput screening technology (*Flaim et al., 2005*; *Brafman et al., 2012*) to engineer in vitro microenvironments that allow for the homogenous expansion of a hPSC-derived mesodermally restricted progenitor population, which we refer to as mesodermal progenitors (MPs). Gene expression analysis and differentiation assays indicated that these cell lack tumor forming potential and exhibit properties associated with intermediate mesoderm, an observation that led us to re-name MP cells as intermediate mesodermal progenitor (IMP) cells. Upon modulation of their culture conditions, IMPs readily generate cell types of the renal lineage. Interestingly, IMP cells fail to differentiate into other mesodermal lineages, such as blood and cardiac muscle. Therefore, IMP cells provide a useful tool to not only study the mechanisms that regulate human mesoderm development but also a homogenous, non-tumorigenic cell source for regenerative medicine purposes.

## Results

### ACME screen to identify culture conditions of MP cells

Using a high-throughput screening platform previously developed in our laboratory referred to as arrayed cellular microenvironments (ACMEs, [*Brafman et al., 2012*]) we sought to identify culture conditions to derive, maintain and expand a cell population with mesodermal properties from hPSCs (including human embryonic and human induced pluripotent stem [hES and iPS] cells). To readily observe and detect acquisition of a mesodermal phenotype, we utilized the hES cell line H9/WA09

harboring the gene encoding green fluorescent protein (GFP) under the control of the *Brachyury* (*T*) promoter (referred to as H9-T-GFP; [*Kita-Matsuo et al., 2009*]). *Brachyury*, which is expressed early in embryonic development in the primitive streak, is transiently expressed as hPSCs exit the pluripotent state and differentiate into mesodermal (ME) lineages (*Rivera-Perez and Magnuson, 2005*).

To induce mesodermal differentiation, we treated H9-T-GFP cells with the GSK3 inhibitor CHIR98014 (CHR) for 2 days, at which point cells uniformly expressed GFP (*Figure 1A*) and were seeded onto ACME slides printed with combinations of bioactive molecules. We performed two sequential screens to identify conditions that maintain GFP expression over a 3-day period: a first screen to identify an optimal substrate composed of extracellular matrix proteins (ECMPs), and a second screen to identify growth factors (GFs) and small molecules (SMs) (*Figure 1A*). The second screen was performed using the optimal substrate composition identified in the first screen. GFP expression for each condition was evaluated and quantified using a high content imaging system and software.

In the first screen, all possible 128 combinations of 7 purified ECMPs (Collagen 1, 3, 4, 5 [C1 C3 C4 C5], Fibronectin [FN], Laminin [LN], Vitronectin [VN]), were tested for their ability to support attachment and maintain GFP expression. Hit conditions were defined as those ECMP combinations that supported maximal cell numbers, as well as GFP expression. The distribution of total cell number and GFP signal intensity across conditions was summarized in a normalized, clustered heat map (*Figure 1B*). Interestingly, several defined ECMP combinations increased total cell number relative to Matrigel, a commercially available extracellular matrix that is commonly used for growth of hPSCs and their derivatives. Further, several ECMP combinations maintained expression of GFP to a greater extent than Matrigel. Cells growing on one of these representative 'hit' conditions (C1 C3 C4 FN VN) is shown in *Figure 1C*.

For the second GF and SM screen, we used one of the optimal matrix compositions (C1 C3 C4 FN VN) as a substrate to deposit combinations of up to three GF and SM, which are known to exert potent effects during early developmental processes. Certain factor combinations increased, while others decreased, cell number and GFP expression (*Figure 1D*). Conditions with positive effects in this assay contained a Wnt agonist (either Wnt3a [WNT] or CHR) and a member of the FGF superfamily (*Figure 1D,E*). Consistent with this observation, a global main effects principal component analysis of all GF and SM revealed that CHR, WNT, Rspondin and FGF exerted the most potent effects on GFP expression (*Figure 1—figure supplement 1*). To a lesser extent, the FGF family members VEGF (VGF) and KGF, also positively influenced GFP expression, whereas Wnt antagonists (DKK1 and IWP2) negatively influenced GFP expression.

We confirmed the ECMP hit conditions by scaling up the 10 top-performing matrix compositions shown in the heatmap of *Figure 1B* into traditional cell culture formats. Compared to Matrigel and a sub-optimal matrix (C1 C4 C5 LN), 8 of the 10 ECMP hit conditions significantly increased the percentage of GFP positive cells (*Figure 2A*). Importantly, in this scaled-up format, the optimal matrix identified in the primary screen (C1 C3 C4 FN VN) consistently led to higher cell numbers and GFP expression compared to the other top ECMP combinations, thus demonstrating the robustness of the ACME screening platform.

We also plated cells in traditional cell culture format on the optimized matrix in the presence of individual soluble factors as well as the top 27 combinatorial hits from the GF-SM screen (*Figure 2B*). This analysis confirmed that the combination of CHR and FGF yielded the highest level of GFP expression and cell number. Since bioactive molecules like CHR (or Wnt) and FGF often exhibit distinct effects at varying concentrations, we performed a dose response analysis to identify optimal CHR and FGF concentrations. The optimal CHR dose was 1.0 μM while the dose of FGF was less dynamic, with its effects saturating at 20 ng/ml FGF (*Figure 2C*).

## Expansion of a mesodermal cell population in defined conditions

The previous analysis was performed 3 days after plating cells in the optimized culture condition. We also examined to what extent this optimized culture condition could support long term growth and expansion of cells with mesodermal properties (*Figure 3A*), which we preliminarily referred to as MP cells. In addition, we extended this analysis to include two additional cell lines, BJ-RiPS and HUES9 cells. When seeded at a density of $10^4$ cells/cm$^2$, cells formed and grew in tight clusters (*Figure 3B*). Cells with these morphological properties were expanded by serial passaging with approximate

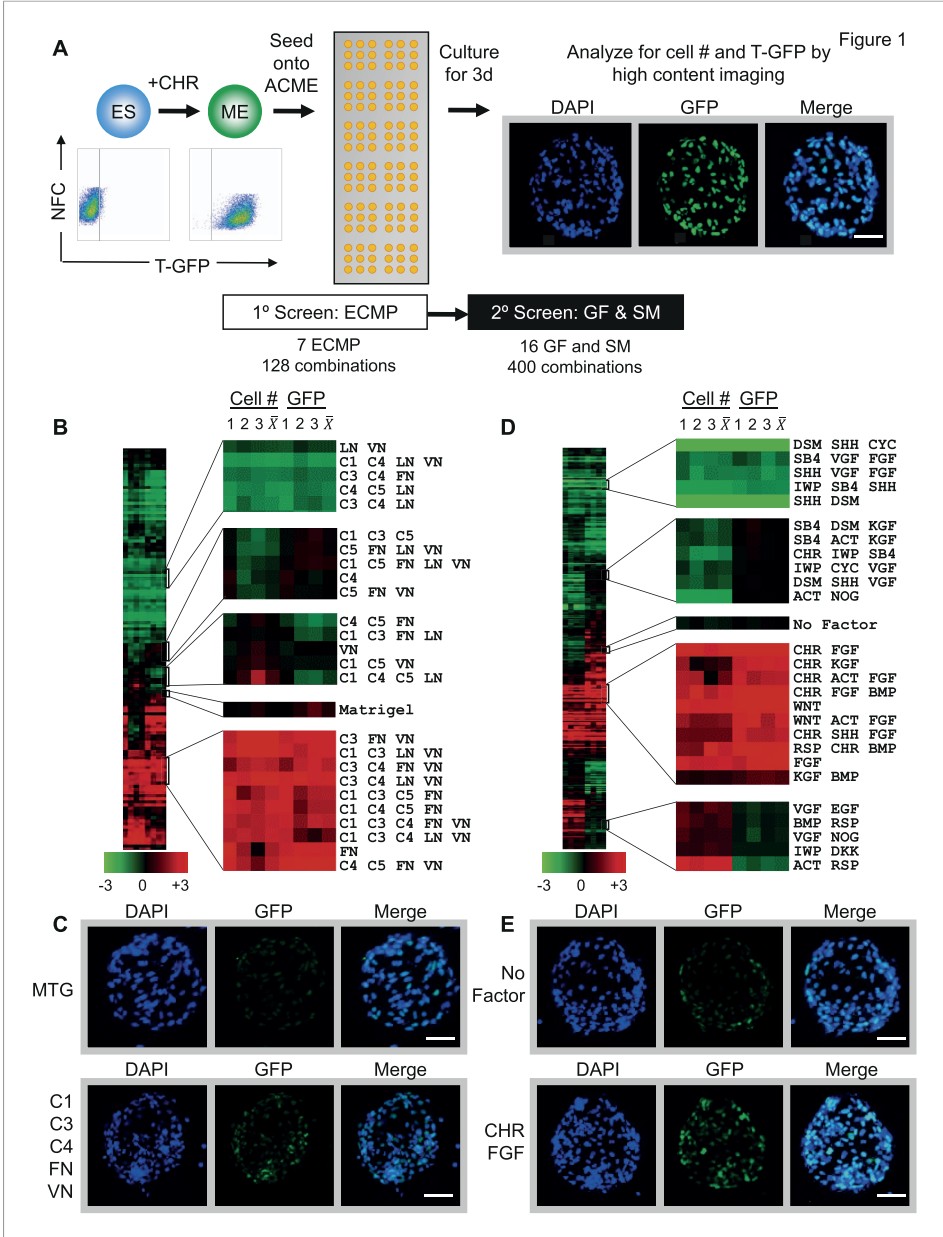

**Figure 1**. Arrayed cellular microenvironment (ACME) screen identified conditions that maintain expression of the mesodermal reporter T-GFP. (**A**) Schematic of the ACME experimental design. Human ES cells carrying a green fluorescent protein (GFP) reporter under control of the BRY/T promoter were treated with CHIR98014 (CHR). GFP positive (T-GFP) cells were seeded onto ACME slides printed with combinations of extracellular matrix proteins (ECMPs), growth factors (GF) and small molecules (SMs). A primary screen contained all possible combinations of ECMP Collagen I (C1), Collagen III (C3), Collagen IV (C4), Collagen V (C5), Fibronectin (FN), Laminin (LN), and Vitronectin (VN). A second GF and SM screen contained all possible single, pairwise, and three-way combinations of Wnt3a (WNT), CHIR98014 (CHR), Rspondin (RSP), Dkk-1 (DKK), IWP-2 (IWP), FGF-2 (FGF), FGF-7 (KGF), VEGF (VGF), EGF (EGF), SHH (SHH), Activin (ACT), Cyclopamine (CYC), Dorsomorphin (DSM), BMP4 (BMP), SB4-31542 (SB4), and Noggin (NOG). The second screen was performed on the optimal ECMP combination identified in the primary screen. 72 hr after seeding, GFP expression and DAPI staining were captured and analyzed using a high content imaging microscope. (**B**) Results of the primary ECMP screen. A heat map of average T-GFP intensity was generated showing the distribution across the data set. Representative clusters are magnified. The position of the Matrigel condition in the cluster is also indicated for reference. Rows represent different ECMP combinations. Columns 1–3 represent biological replicates for cell number (Cell #) or T-GFP (GFP). Columns marked $\overline{X}$ represent the average of the three biological replicates. (**C**) Representative images of ECMP conditions in the array format. Matrigel is shown

*Figure 1. continued on next page*

*Figure 1. Continued*

in comparison to the hit condition C1 C3 C4 FN VN. Scalebar = 50 µm. (**D**) Results of the second GF and SM screen. A heat map of average T-GFP intensity was generated showing the distribution across the data set. Representative clusters are magnified. The position of the condition lacking GFs and SMs (No Factor) is also indicated for reference. Rows represent different GF and SM combinations. Columns 1–3 represent biological replicates for cell number (Cell #) or T-GFP (GFP). Columns marked $\overline{X}$ represent the average of the three biological replicates. (**E**) Representative images of GF and SM conditions in the array format. No GF or SM is shown in comparison to the hit condition CHR + FGF. Scalebar = 50 µm. *Figure 1—figure supplement 1* provides a global main effects principal component analysis for all GF and SM used in this second screen.

The following figure supplement is available for figure 1:

**Figure supplement 1**. Global main effects principal component analysis of GF and SM ACME screen demonstrates that WNT and FGF agonists exert positive effects on T-GFP expression.

doubling rates of 60.2 ± 4.2 hr (H9 = 55.4 hr, *Figure 3C*; Hues9 = 61.8 hr, RiPS = 63.4 hr, *Figure 3—figure supplement 1*) and expressed the proliferation marker Ki-67 (*Figure 3—figure supplement 2*). Cell counts taken at each passage revealed that $1 \times 10^4$ cells could theoretically be expanded to approximately $1 \times 10^{12}$ cells over 10 passages (*Figure 3C* and *Figure 3—figure supplement 1*). These cells maintained 46 chromosomes (*Figure 3—figure supplement 3*), indicating that cultured cells did not acquire abnormal chromosome numbers commonly associated with late passage hPSCs. Reverse transcription quantitative PCR (qPCR) showed that expression of genes associated with pluripotency (*OCT4, NANOG, SOX2*) was rapidly lost during expansion (*Figure 3D*; *Figure 3—figure supplement 4*). This loss of pluripotency-associated properties was further confirmed by immunofluorescence (IF) staining (OCT4 and NANOG, *Figure 3—figure supplement 5*) and flow cytometry (TRA-1-81 and SSEA4, *Figure 3—figure supplement 6*). In contrast, genes associated with the mesoderm (ME) lineage (*MESP1, MIXL1, LHX1*) were upregulated and maintained over 10 passages (*Figure 3E*; *Figure 3—figure supplement 7*). IF staining confirmed the presence of MIXL1 protein in these expanded cell cultures (*Figure 3F*). Using flow cytometry, we furthermore showed that the expanded cells shared a cell surface signature of CD56+ CD326– (*Figure 3G*; *Figure 3—figure supplement 8*), previously defined for a multipotent mesoderm-committed cell population (*Evseenko et al., 2010*). In addition, expression of the EN marker *FOXA2* and the EC marker *SOX1* was significantly reduced in these cells (*Figure 3H,I*; *Figure 3—figure supplements 9, 10*).

The apparent indefinite expansion of MP cells (greater than 20 passages at the time of this submission) raised the possibility that these cells, like undifferentiated hPSCs, harbored tumorigenic potential. Importantly, unlike hPSCs, MP cells did not produce tumors when injected into immune compromised mice (*Figure 3J*). Among the 12 MP cell injections (2 injections per mouse ranging from 0.5 million to 1 million cells), only one site maintained a small lump (~1 mm in diameter), which did not grow in size over 12 weeks. In contrast, all 6 hPSC injections (0.5 million cells per injection) produced readily visible teratomas (greater than 10 mm in diameter). Taken together, we have generated a non-tumorigenic progenitor population capable of nearly indefinite expansion potential with a mesodermal phenotype.

## Optimized culture conditions are necessary to generate and maintain MP cells

From the ACME screens, we identified a defined matrix (C1 C3 C4 FN VN) and combination of soluble factors (CHR + FGF) that allow for the derivation and expansion of MP cells. We wanted to explore to what extent these defined conditions were critical for the derivation and expansion of MP cells. To this end, we first compared the effectiveness of our defined matrix relative to Matrigel and of CHR + FGF relative to no factors in deriving MP cells (*Figure 4A*), as assayed by qPCR of mesodermal markers. Importantly, cells cultured in the absence of CHR and/or FGF failed to passage beyond one passage, indicating that these soluble factors are essential to the expansion of MP cells. Furthermore, although Matrigel with CHR and FGF yielded cells expressing the mesodermal markers *MESP1, MIXL1*, and

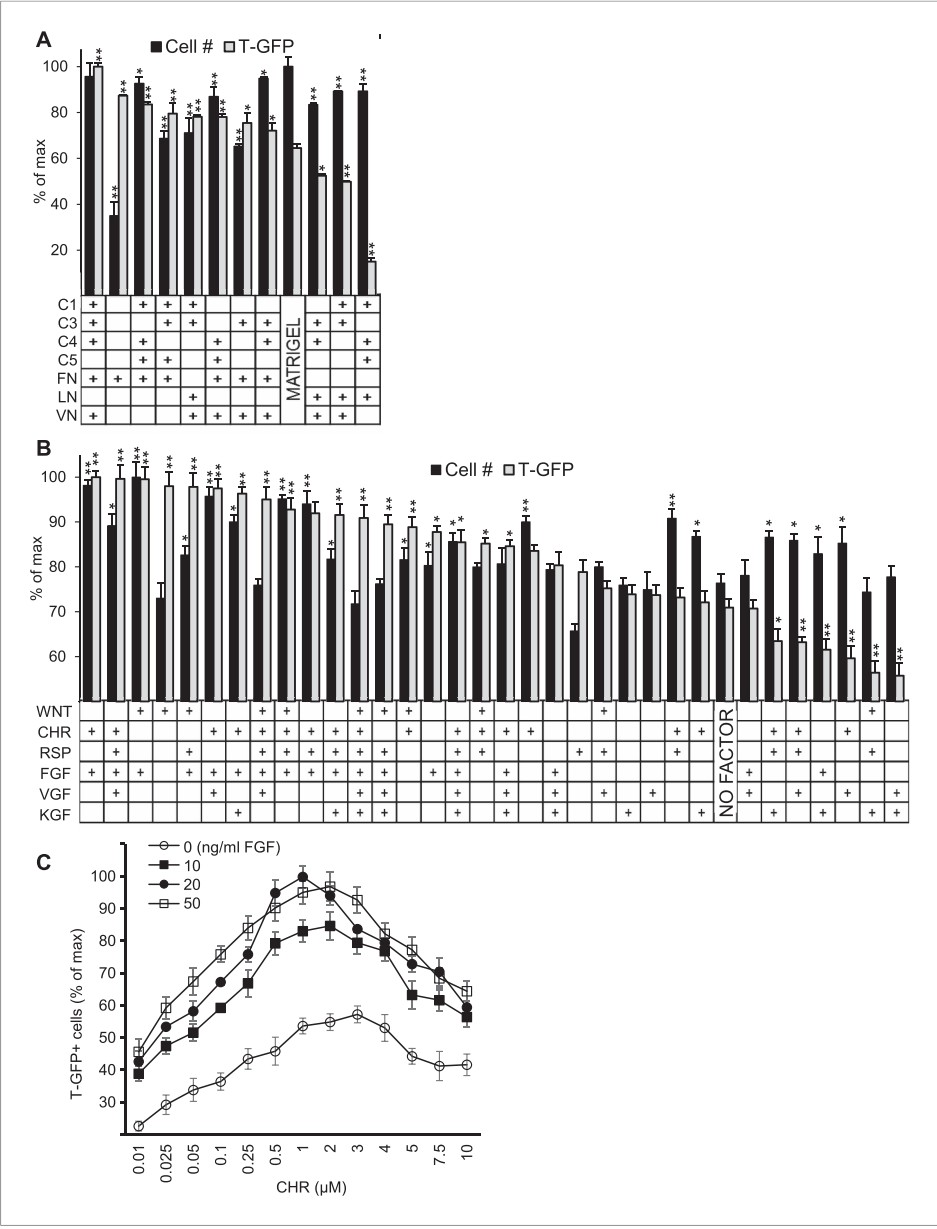

**Figure 2**. Validation of high-throughput ACME screens. Scale up analysis of hits from the ACME screens. Human ES cells carrying a GFP reporter under control of the BRY/T promoter were treated with CHIR98014 (CHR) for 24 hr. After 48 hr, GFP positive (T-GFP) cells were cultured in multi-well plates for 72 hr to validate conditions from the ACME screens. (**A**) GFP+ cells were cultured in multi-well plates coated with 10 hit matrices from the primary ECMP screen as well as Matrigel and a sub-optimal matrix (C1 C4 C5 LN). The optimal matrix (C1 C3 C4 FN VN) was defined as the condition that maintained the highest T-GFP expression and fostered the highest cell number. Statistical comparisons are made to the Matrigel condition. *p < 0.05, **p < 0.005. When p-values are not indicated with * or **, the statistical difference is not significant from the control. (**B**) GFP+ cells were cultured in multi-well plates coated with the optimal matrix (C1 C3 C4 FN VN) and various GF/SM combinations. Statistical comparisons are made to the conditions containing no GF/SM (No Factor). *p < 0.05, **p < 0.005. When p-values are not indicated with * or **, the statistical difference is not significant from the control. (**C**) GFP+ cells were cultured in multi-well plates coated with the optimal matrix (C1 C3 C4 FN VN) and various concentrations of CHR and FGF2 (FGF).

*LHX1*, our optimized matrix significantly increased their expression (***Figure 4B***). By passage 3, cells cultured in our optimized conditions expressed 1.5- to 2-fold greater levels of *MESP1*, *MIXL1*, and *LHX1* compared to cells cultured on Matrigel (***Figure 4B***).

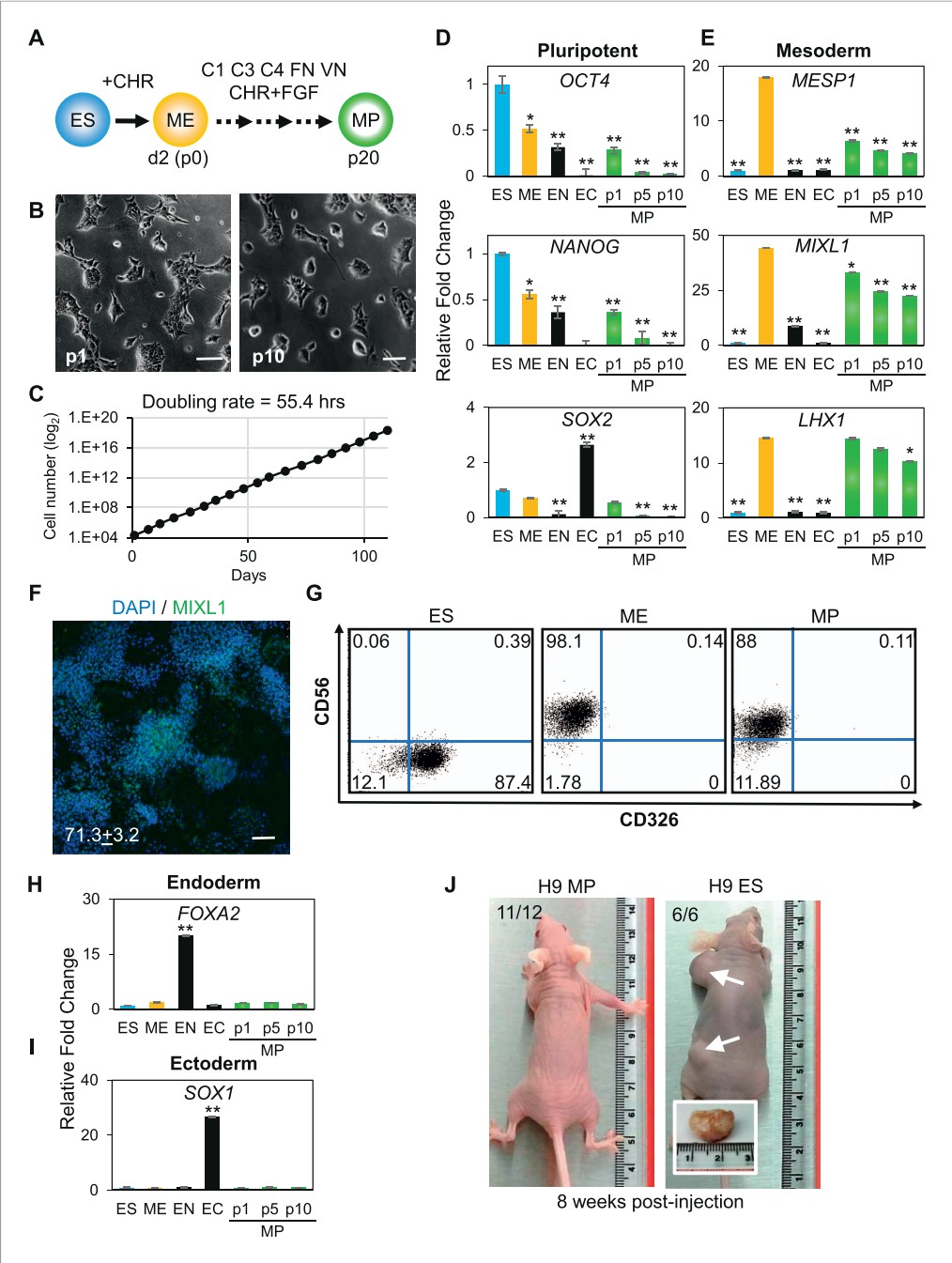

**Figure 3**. Characterization of mesodermal progenitor population. (**A**) Schematic showing derivation of mesoderm progenitor (MP) cells. Human ES cells were differentiated into mesoderm (ME) with CHIR98014 (CHR) and then replated onto the defined substrate C1 C3 C4 FN VN and cultured with CHR and FGF2 (FGF) for up to 20 passages (p0 to p20). (**B**) Representative images of MP cells derived from the hES cell line H9/WA09 at passage 1 and 10 in C1 C3 C4 FN VN with CHR and FGF. Scale bar = 50 μm. (**C**) Growth rate of MP cells derived from H9 T-GFP. Cell counts were taken at each passage. (**D**) Quantitative PCR (qPCR) analysis for expression of pluripotency markers *OCT4*, *NANOG*, and *SOX2*. Expression of these markers in MP cells at passages 1, 5 and 10 is lower than in undifferentiated cells (ES). Cells differentiated into ME, endoderm (EN) and ectoderm (EC) served as controls. All statistical comparisons are made to the ES sample. *p < 0.05, **p < 0.005. (**E**) qPCR analysis for expression of mesodermal markers *MESP1*, *MIXL1*, and *LHX1*. Expression of these markers in MP cells at passages 1, 5 and 10 is comparable to that observed in ME and higher than in ES, EN and EC. All statistical comparisons are made to the ME sample. *p < 0.05, **p < 0.005. (**F**) MIXL1 immunofluorescence (IF) in MP cells. MP cells at passage 15 were fixed and stained with MIXL1-specific antibody. Number indicates percentage of MIXL1 expressing cells in the MP cell population.
*Figure 3. continued on next page*

*Figure 3. Continued*

Standard deviation represents the variation between the fields of view used for counting (n = 20). Scale bar = 50 μm. (**G**) Flow cytometry analysis for CD56 (NCAM) and CD326 (ECAM). Pluripotent cells (ES, CD326+ CD56−) are differentiated to ME cells (CD326− CD56+). MP cells at p10 exhibit a similar cell surface expression of these two markers as ME. (**H**) qPCR analysis for expression of the EN marker *FOXA2*. Expression of *FOXA2* is only detected in cells differentiated towards EN. All statistical comparisons are made to the ES sample. (**I**) qPCR analysis for expression of the EC marker *SOX1*. Expression of *SOX1* is only detected in cells differentiated towards EC. All statistical comparisons are made to the ES sample. (**J**) MP cells are non-tumorogenic. Nude mice were injected with H9-derived MP cells or H9 ES cells. Injected ES cells generated tumors while injected MP cells did not form any growth in 11/12 injections. *Figure 3—figure supplement 1* through 10 provide additional analysis, including for two other hPSC lines (BJ RiPS and HUES9).

The following figure supplements are available for figure 3:

**Figure supplement 1**. Growth rate of MP cells derived from Hues 9 or BJ RiPS.

**Figure supplement 2**. Flow cytometry analysis of Ki-67 in human ES, ME, and MP.

**Figure supplement 3**. Karyotype of MP cells derived from the hES cell line H9/WA09.

**Figure supplement 4**. QPCR analysis for expression of pluripotency markers *OCT4*, *NANOG*, and *SOX2*.

**Figure supplement 5**. IF of Hues 9 ES and MP cells demonstrate that MP cells do not express OCT4 and NANOG proteins.

**Figure supplement 6**. Flow cytometry analysis of Hues 9 ES and MP (p10) cells for Tra-1-81 and SSEA4.

**Figure supplement 7**. QPCR analysis of MP cells derived from Hues 9 and BJ RiPS for expression of mesodermal markers *MESP1*, *MIXL1*, and *LHX1*.

**Figure supplement 8**. Flow cytometry analysis for CD56 (NCAM1) and CD326 (EPCAM) in undifferentiated RiPS cells as well as ME and MP (p10) cells derived from RiPS cells.

**Figure supplement 9**. QPCR analysis of MP cells derived from Hues 9 and BJ RiPS.

**Figure supplement 10**. QPCR analysis of MP cells derived from Hues 9 and BJ RiPS.

---

Next, we compared the effectiveness of our defined matrix relative to Matrigel and of CHR + FGF relative to no factors in maintaining MP cells (*Figure 4C*). For this analysis, MP cultures were grown in the optimized conditions (C1 C3 C4 FN VN and CHR + FGF) through passage 6, at which point cultures were either passaged onto Matrigel or the defined matrix in the presence or absence of the soluble factors CHR and FGF. Again, the optimized culture condition produced a statistically significant difference in maintaining mesoderm marker expression compared to other conditions (*Figure 4D*). Importantly, MP cultures without CHR and FGF failed to expand beyond the first passage. Taken together, these results indicate that the defined substrate C1 C3 C4 FN VN as well as CHR and FGF are required for optimal MP cell generation and maintenance.

## Global gene expression demonstrates an intermediate mesodermal (IM) identity of MP cells

To further characterize the MP cell population derived and expanded under our defined culture conditions, we performed transcriptome analysis by RNA sequencing (RNA-seq). For comparison, we analyzed the transcriptomes of undifferentiated hES cells, as well as of transient EC, EN, ME populations differentiated from hES cells. Cluster analysis of the RNA-seq data revealed that MP cells are more similar to ME cells than they are to EC, EN, and hES cells (*Figure 5A* and *Supplementary file 1A,B*). Comparison of expressed genes in MP and ME cell populations confirmed a high degree of similarity,

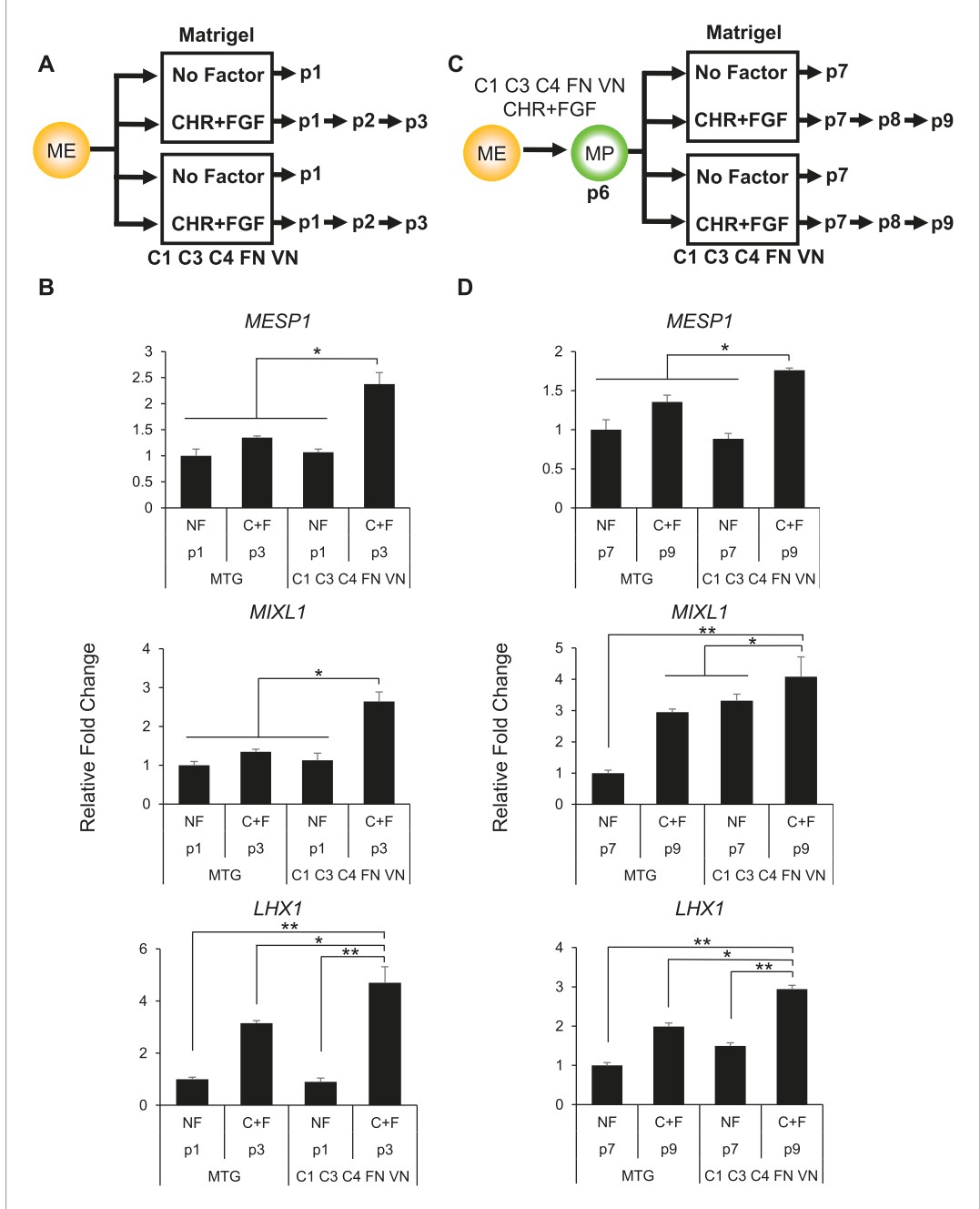

**Figure 4**. Optimized culture conditions are required to generate and maintain MP cells. (**A**) Human ES cells were treated with CHIR98014 (CHR) for 24 hr. After 48 hr, cells were cultured on either Matrigel or the optimal matrix (C1 C3 C4 FN VN) in the absence (no factor) or in the presence of the optimal GF/SM combination (CHR + FGF). Only cells cultured with CHR + FGF could be serially passaged. (**B**) QPCR analysis for mesodermal markers *MESP1*, *MIXL1*, and *LHX1*. Conditions containing no factor did not grow beyond passage 1, while the CHIR + FGF samples represent expression at passage 3. NF = no factor; C + F = CHR + FGF. Statistical comparisons are made to C1 C3 C4 FN VN with CHR + FGF condition. *p < 0.05, **p < 0.005. (**C**) MP cells were expanded to p6 on the optimal ECMP (C1 C3 C4 FN VN) and GF/SM combination (CHR + FGF). MP cells were then either transitioned to Matrigel or maintained on C1 C3 C4 FN VN in the absence or presence of CHR + FGF. (**D**) QPCR analysis for mesodermal markers *MESP1*, *MIXL1*, and *LHX1*. Conditions containing no factor did not grow past p7, while the CHR + FGF sample represents expression at p9. All statistical comparisons are made to the C1 C3 C4 FN VN with CHR + FGF condition. *p < 0.05, **p < 0.005.

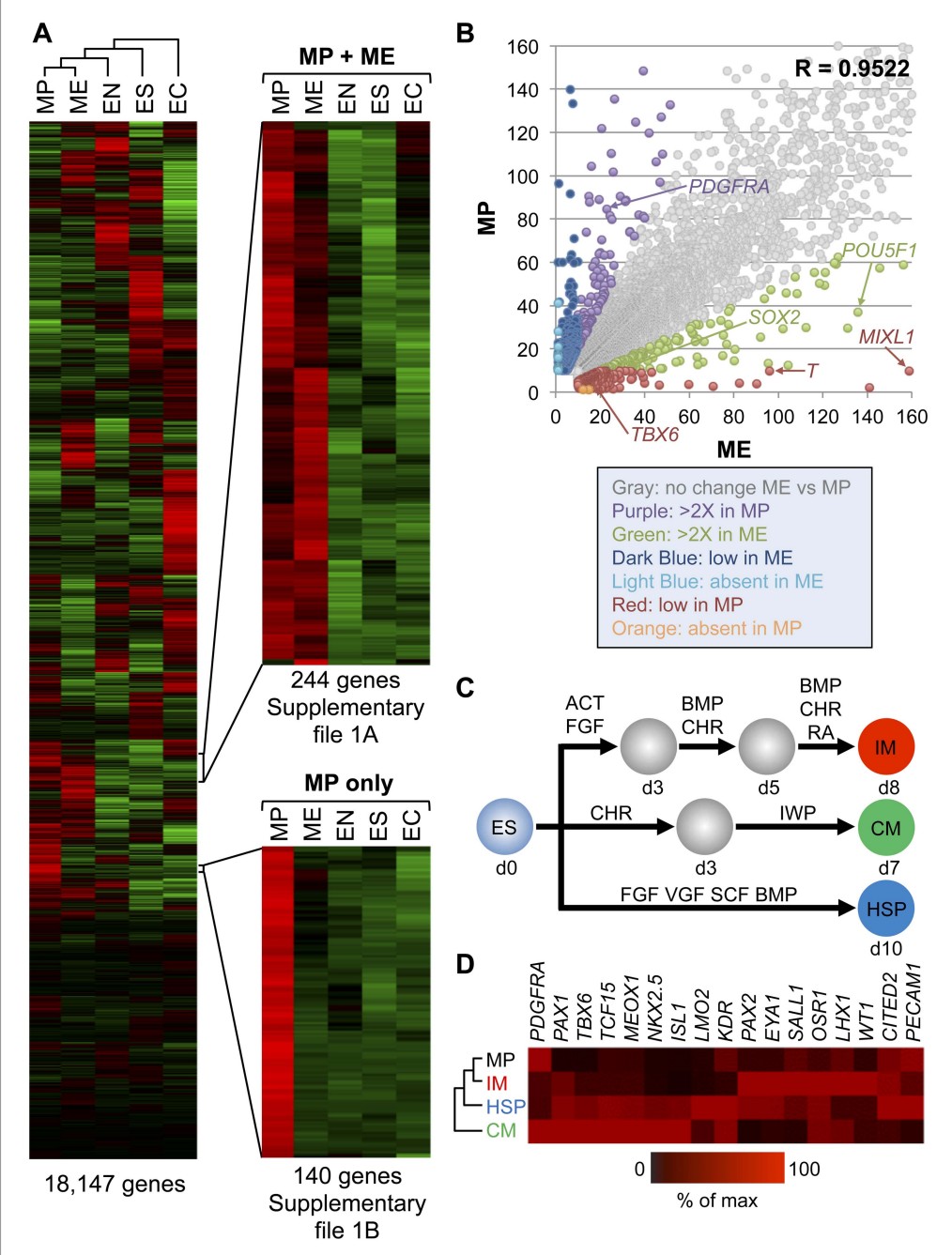

**Figure 5**. Gene expression analysis reveals that MP cells have an intermediate mesodermal (IM) identity. RNA sequencing (RNA-seq) was used to analyze gene expression of MP cells. As a comparison, gene expression profiles were analyzed for hES (ES) cells and their differentiated progeny, ME, EN and EC. (**A**) MP cells resemble mesodermally differentiated cells. Hierarchical clustering analysis was performed for all genes with detectable expression (reads per kilobase per million mapped reads [RPKM] values greater than 10) in one of the five cell populations. *Supplementary file 1* provides the complete list of genes shared between MP and ME (**A**) and genes unique to MP (**B**). The complete RNA-seq data set for MP cells is provided in *Supplementary file 2*. (**B**) Correlation of gene expression profiles. Genes with expression values (RPKM) expression between 10 and 1500 were plotted for MP cells and ME. The correlation coefficient (R) for all expressed genes is 0.9522. (**C**) Schematic depicting differentiation protocols from hES cells to IM and lateral plate mesoderm (LM) derivatives cardiomyocytes (CMs) and hematopoietic stem and progenitor (HSP) cells. (**D**) QPCR analysis of IM, CM, HSP, and MP cells revealed that MP cells have a similar expression profile as IM cells. ACT = Activin A, BMP = BMP4, CHR = CHIR98014, d = day, FGF = FGF2, IWP = IWP-2, RA = retinoic acid, VGF = VEGF.

with a correlation coefficient of 0.9522 (*Figure 5B*). Although this analysis revealed that MP cells are more similar to transient ME populations than they are to other cell populations examined, they are also distinct from ME cells. In contrast to ME cells, MP cells exhibit significantly lower levels of pluripotency regulators, including *POU5F1* (*OCT4*) and *SOX2*. Several established early mesodermal markers (*T*, *MIXL*) were significantly elevated in ME cells relative to MP cells, suggesting that MP cells have progressed beyond this transient and early ME phenotype.

During development as the ME germ layer matures, modulation of various signaling molecule pathways lead to its further specification into paraxial, intermediate, and lateral plate mesoderm (PM, IM, and LM, respectively) (reviewed in *Christ and Ordahl, 1995*). IM develops into cells of the urogenital system, whereas LM develops into tissues of the vascular system, including cardiomyocytes (CMs) and hematopoietic stem and progenitor (HSP) cells. Using established differentiation protocols (*Figure 5C*), we examined expression by qPCR of several mesodermal markers in MP cells relative to IM, CM and HSP. Interestingly, MP cells most closely resembled the mesodermal gene expression profile of IM cells (*Figure 5D*). In addition, we observed in the RNA-seq data that several IM markers (*CITED2*, *EYA1*, *GATA3*, *LHX1*, *SALL1*) were expressed in MP cells (*Supplementary file 2*). Based on this gene expression analysis we speculated that MP cells are most closely related to cells of intermediate mesoderm and consequently renamed them from MP cells to intermediate mesodermal progenitor cells (IMP).

## IMP cells are restricted to differentiate towards an intermediate mesoderm phenotype

Based on the above findings, we hypothesized that the differentiation potential of IMP cells may be limited to cell types derived from IM, such as of the renal lineage. To test this hypothesis, we tested the ability of IMP cells to differentiate into various mesodermally derived tissues, including hematopoietic cells, CMs and renal progenitors. Using an established protocol for hematopoietic differentiation (*Figure 6A*, adapted from *Ng et al. (2008)*), we successfully differentiated hES cells into cells expressing *SOX17*, a marker of hemogenic endothelium, and CD34 and CD45, two cell surface markers commonly used to monitor the presence of hematopoietic cell populations (*Figure 6B,C*). In contrast, IMP cells derived from three independent hPSC lines and manipulated in a similar manner failed to express these markers at detectable levels (*Figure 6B,C*), even when culture periods were extended beyond the standard protocol.

Along similar lines, using an established protocol to derive CMs (*Figure 6D*, adapted from *Lian et al. (2012)*), hES cells readily produced cardiac progenitors (CPs) and subsequently CMs, as monitored by expression of *NKX2.5* and *ISL1* (*Figure 6E*). Cultures containing CMs exhibited the characteristic contractile activity associated with such cells. In contrast, IMP cells subjected to these same manipulations failed to express detectable levels of *NKX2.5* and *ISL1* (*Figure 6E*), and never produced contractile activity. Furthermore, since CM differentiation from hES cells is enhanced by inhibition of Wnt signaling (*Willems et al., 2011*), we reasoned that a prolonged withdrawal of CHR (a potent Wnt agonist required to maintain IMP cells) and addition of IWP (a potent Wnt inhibitor) may encourage IMP cells to enter the CM lineage. However, under no tested conditions were we able to promote CM differentiation from IMP cells. Taken together, IMP cells were unable to differentiate into cells with hematopoietic or cardiogenic properties, both derivatives of LM.

Since the IMP cells described in this study failed to generate derivatives of LM, we reasoned that these cells may differentiate into cell populations derived from IM, such as kidney and gonads. To test this possibility, we employed a published protocol to differentiate hES cells into renal progenitors (*Figure 7A*) (*Taguchi et al., 2014*). This protocol employed several GFs and SMs to promote the differentiation of hES cells to IM and subsequently metanephric mesenchyme (MM). Importantly, IMP cells efficiently acquired gene expression signatures associated with IM and MM as monitored by qPCR (*Figure 7B*). The gene expression profile of IMP-derived MM exhibited a striking similarity to that of fetal kidney cells. *PAX2* and *SIX2* were upregulated at day 14 of renal differentiation, indicating commitment to the kidney lineage (*Bush et al., 2013*). Furthermore, immuno-fluorescence analysis demonstrated that a significant number of cells expressed IM and MM markers PAX2, SALL1, SIX2, WT1 and CDH1 (E-cadherin) (*Figure 7C–E*). These results suggested that IMP cells, as predicted by the gene expression profile, are restricted to IM and effectively differentiate into cells expressing genes associated with a renal phenotype.

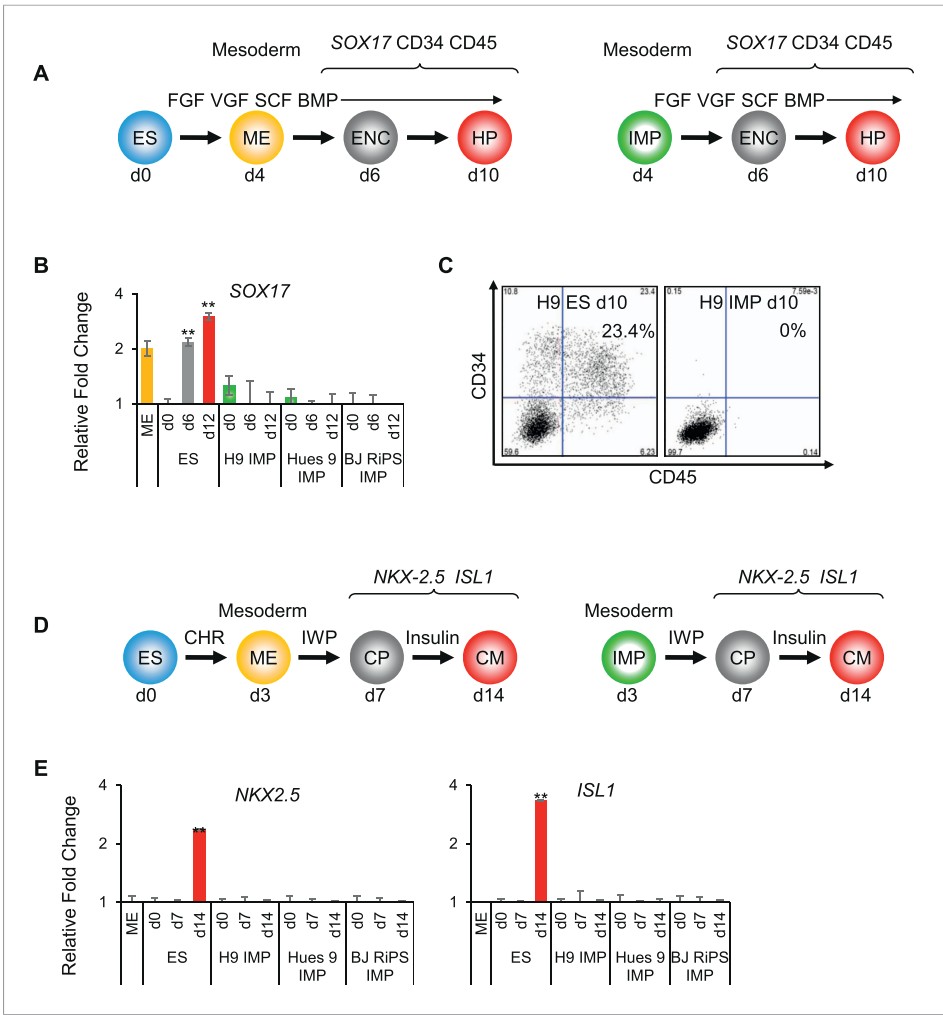

**Figure 6**. IMP cells are unable to differentiate to cell types derived from lateral plate mesoderm (LM). (**A**) Schematic of the hematopoietic differentiation protocol. Cells were differentiated in a step-wise manner using the indicated GFs and SMs from undifferentiated ES cells or from IMP cells to ME, endothelial cell (ENC) and subsequently to hematopoietic precursors (HPs). Stage-specific marker genes and cell surface markers expressed during this differentiation process are indicated at the top. FGF = FGF2, VGF = VEGF, SCF = Stem Cell Factor, BMP = BMP4. (**B**) QPCR analysis of hES and MP cells differentiated towards HPs. Compared to hES cells, IMP cells do not differentiate towards HPs, as indicated by the absence of *SOX17* expression. (**C**) Flow cytometry analysis of hES and IMP cells differentiated towards HPs for CD34 and CD45. While hESC cells can differentiate into CD34+ CD35+ HPs, IMP cells fail to differentiate generate cells positive for CD34 and CD45. (**D**) Schematic of the CM differentiation protocol. Cells were differentiated in a step-wise manner using the indicated GFs and SMs from undifferentiated ES cells or from IMP cells to ME, cardiac progenitor (CP) and subsequently to CM. Stage-specific marker genes expressed during this differentiation process are indicated at the top. CHR = CHIR98014, IWP = IWP-2. (**E**) QPCR analysis of MP cells differentiated towards CMs. Compared to hES cells, IMP cells do not differentiate towards CMs, as indicated by the absence of *ISL1* and *NKX2.5* expression.

To further assess the ability of the IMP cells to generate cells with renal properties, we employed two rat explant assays that represent stringent measures of renal potential. In the first assay, we co-cultured IMP-derived MM cells with dissected embryonic rat spinal cords (SCs), a tissue that produces potent nephrogenic inductive signals (*Figure 8A*) (*Kispert et al., 1998*; *Osafune et al., 2006*; *Gallegos et al., 2012*). In this system, IMP-derived MM cells readily acquired expression of markers associated with renal cell types, including Lotus tetragonolobus lectin (LTL), CDH1, SALL1 and SIX2 (*Figure 8B*). In contrast, undifferentiated hES cells failed to express of SIX2 (*Figure 8C*), indicating that MM properties are required for efficient renal differentiation. Although IMP-derived

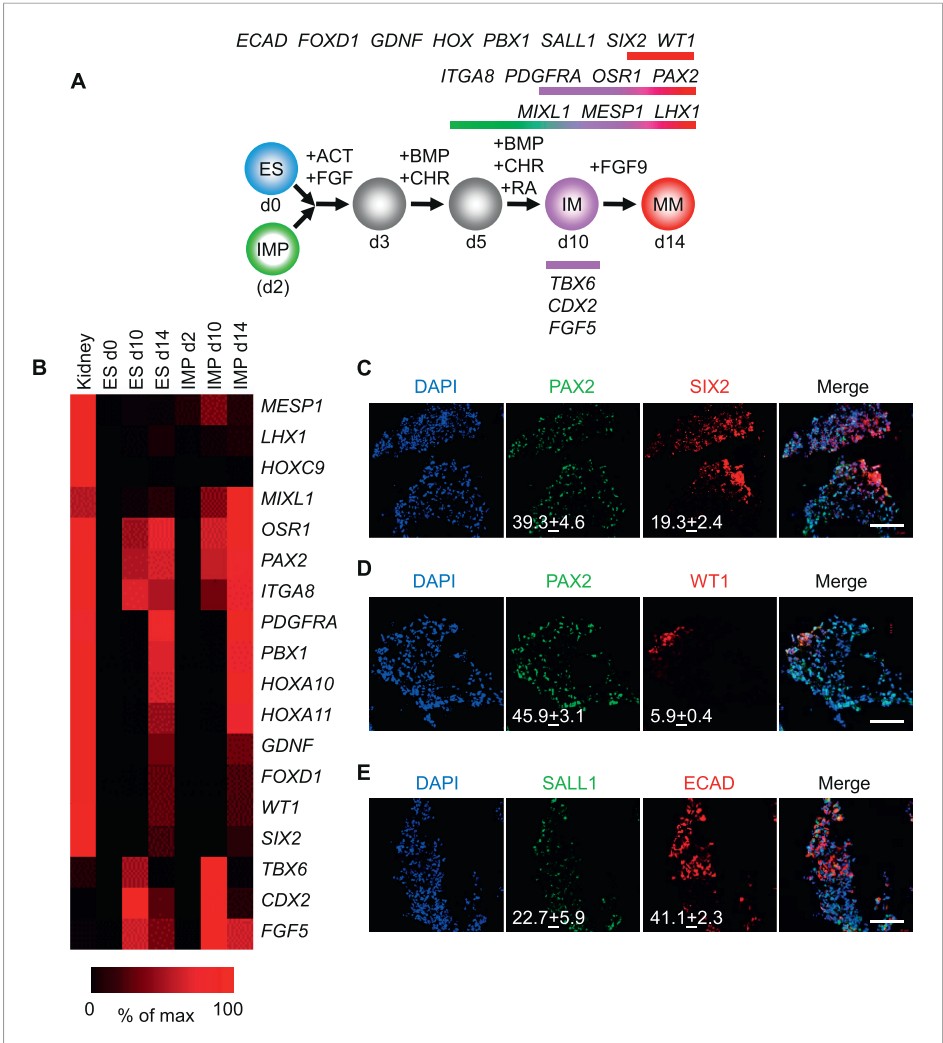

**Figure 7**. Differentiation of IMP cells into metanephric mesenchyme (MM). (**A**) Schematic of the differentiation protocol. Cells were differentiated in a step-wise manner using the indicated GFs and SMs from undifferentiated ES cells or from IMP cells to IM and subsequently to MM. Stage-specific marker genes expressed during this differentiation process are indicated at the top. ACT = ActivinA, BMP = BMP4, CHR = CHIR98014, d = day, FGF = FGF2, RA = retinoic acid. (**B**) Upon differentiation towards MM, cells expressed genes associated with kidney lineage. QPCR was performed on ES and IMP cells for the indicated genes at various time points. Fetal kidney RNA (11 gestation weeks) was used as a control. The data is displayed as a heat map with black corresponding to minimal expression and red corresponding to maximal levels. (**C–E**) IF analysis of MP cell-derived MM. IMP cells were differentiated as depicted in panel **A**, fixed and stained for the indicated proteins and DNA (DAPI). Numbers refer to percentages of cells expressing the protein of interest. Standard deviation represents the variation between the fields of view used for counting (n = 20). Scale bar = 100 μm.

MM cells expressed several markers associated with the renal lineage, they failed to generate tubule-like structures, including the nephron, suggesting that IMP cells differentiate effectively into a sub-population of kidney cells. These co-culture experiments demonstrate that IMP cells efficiently generate cell types with renal characteristics.

In a second assay, rat embryonic kidneys were dissociated to single cells and re-aggregated to form kidney-like organoids (*Unbekandt and Davies, 2010*; *Davies and Chang, 2014*). These aggregation experiments were performed in the presence of either IMP-derived MM cells (*Figure 9A*) or undifferentiated hES cells (control), thereby assessing the renal potential of these cells. The contribution of human cells to the re-aggregated rat kidneys is readily detected by staining for the

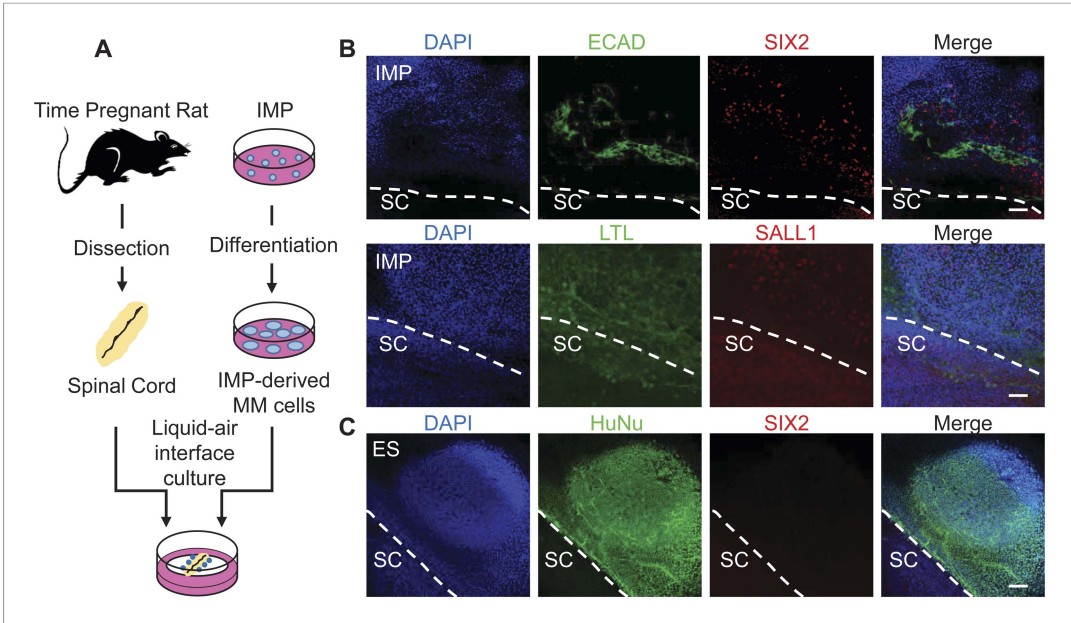

**Figure 8**. Assessment of renal potential of IMP cells. (**A**) Schematic of spinal cord (SC) co-culture assay to assess renal differentiation potential of IMP cells. IMP cells were differentiated as depicted in *Figure 7A* and incubated in liquid–air interface cultures with rat embryonic SC explants. (**B**) Immuno-fluorescence analysis of markers expressed in renal progenitors. 4 days after co-cultures were established, cells were fixed and stained for the indicated proteins (ECAD, SIX2 and SALL1) and for Lotus-tetragonolobus lectin (LTL). The dashed line indicates the boundary between human cells and the SC explant. Scale bar = 100 μm. (**C**) Undifferentiated hES cells failed to express SIX2 when co-cultured with embryonic rat SCs. Scale bar = 100 μm.

human specific nuclear antigen (HuNu). In this assay, we consistently observed efficient incorporation of IMP-derived MM cells into the kidney organoids (*Figure 9B*, *Figure 9—figure supplement 1*). Interestingly, we primarily observed incorporation of these cells into the mesenchyme surrounding epithelial structures, which were visualized by staining with lectin Dolichos biflorus agglutinin (DBA). Furthermore, incorporated human cells expressed FOXD1, the expression of which is restricted to metanephric stromal mesenchyme (*Hatini et al., 1996*) and stained with LTL (*Figure 9C*, *Figure 9—figure supplement 2*). In contrast, undifferentiated hES cells failed to incorporate into these kidney organoids (*Figure 9D*, *Figure 9—figure supplement 3*) and instead were found adjacent to the organoid structures (*Figure 9—figure supplement 3*, bottom row). Taken together, these co-culture experiments establish that IMP cells efficiently incorporated into the developing kidney.

## Discussion

In this study, we describe a novel progenitor cell population derived from hPSCs with the potential to differentiate into tissues of the IM lineage. By using the ACME screening technology, we were able to simultaneously define and optimize derivation and expansion conditions for these intermediate mesodermal progenitor (IMP) cells. Although it was our initial intention to produce a progenitor cell population with broad differentiation potential into all mesodermally-derived tissues, we made the surprising finding that the differentiation potential of these IMP cells was restricted to the IM lineage. Consequently, we were unable to coax IMP cells to differentiate into cell types derived from LM, such as blood and CMs. This exquisite lineage restriction was particularly surprising in light of the expression of multiple pan-mesodermal marker genes, such as *LHX1*, *MESP1* and *MIXL1*. Given their ability to differentiate into cell types with gene expression patterns associated with renal lineages, we hypothesize that this IMP cell population is an in vitro counterpart to intermediate mesoderm. It will be interesting to investigate whether IMP cells are capable of differentiating into other derivatives of intermediate mesoderm, such as the Wolffian and Müllerian ducts of the developing reproductive system.

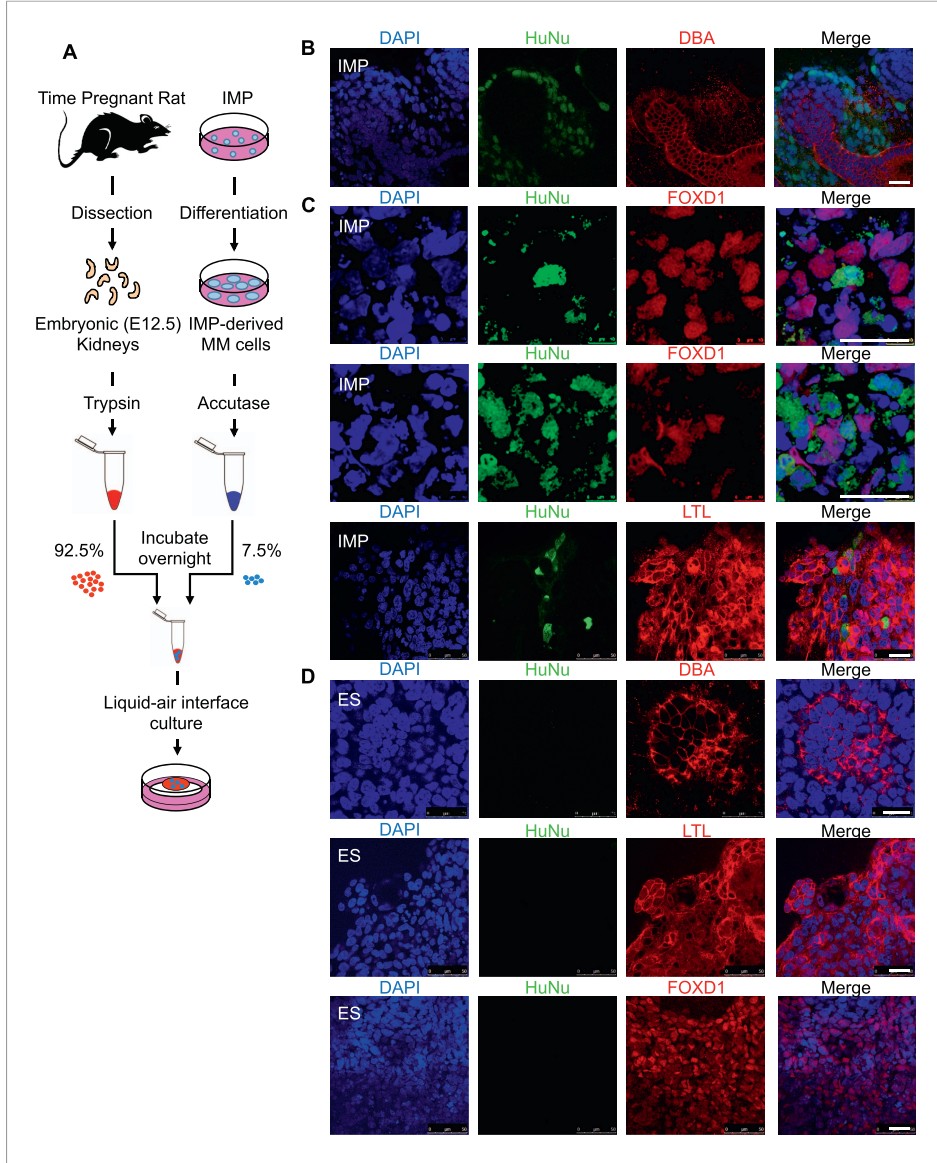

**Figure 9**. Incorporation of IMP cells into kidney mesenchyme. (**A**) Schematic of a re-aggregation assay to test renal potential. IMP cells were differentiated as depicted in *Figure 7A* and mixed with dissociated embryonic rat kidneys at a ratio of 7.5:92.5 and co-incubated for 4 days to form organoids in media-air interface co-culture.
(**B**) Representative images of re-aggregated kidney organoids. IMP cells differentiated to MM are detected with the human specific nuclear antigen HuNu (green). Human cells are clearly integrated into renal organoids and surround epithelial structures labeled with the lectin Dolichos biflorus agglutinin (DBA) (red). *Figure 9—figure supplement 1* provides additional images of MP cells incorporating into renal structures. Scale bar = 25 μm. (**C**) Representative images of re-aggregated kidney organoids. Renal organoids were labeled with DAPI (blue) to identify nuclei, HuNu (green) to identify human cells and with either FOXD1 antibody or LTL (red). Two representative sets of images are shown to indicate co-localization of FOXD1 in HuNu positive cells. Scale bar = 25 μm. (**D**) Undifferentiated hES cells failed to integrate into renal organoids. Instead of MP cells, undifferentiated ES cells were mixed with dissociated embryonic rat kidneys. These cells failed to integrate into the renal organoid structures as indicated by the lack of HuNu staining. *Figure 9—figure supplement 2* demonstrates that undifferentiated ES cells fail to incorporate into these structures. Scale bar = 25 μm.

The following figure supplements are available for figure 9:

**Figure supplement 1**. Additional assessment of renal potential of MP cells.

*Figure 9. continued on next page*

*Figure 9. Continued*

**Figure supplement 2**. Staining controls relevant to *Figure 9C*.
**Figure supplement 3**. Additional assessment of renal potential of MP cells.

Generation of expandable, lineage restricted progenitor cell populations offers several advantages over the use of undifferentiated hPSCs in tissue engineering approaches. First, differentiated cultures derived directly from hPSCs often harbor undifferentiated cells, which retain the potential to seed tumor growth. Such tumor-initiating potential is problematic when cells are intended for transplantation to repair or replace damaged tissue. Based on our sub-cutaneous injections into immune-compromised mice, IMP cells do not grow into teratomas, a defining property of undifferentiated pluripotent stem cells. Our gene expression analysis provides further evidence of this loss of pluripotency and hence of teratoma-seeding potential: IMP cells express nearly undetectable levels of pluripotency markers, such as *POU5F1/OCT4* and *SOX2*, both of which show residual expression in mesodermally differentiated hPSCs. Second, lineage-restricted progenitors require less elaborate manipulation to derive more mature cell populations. In the case of the IMP cells, early differentiation steps to usher cells into a mesodermal lineage are no longer needed, thereby truncating differentiation protocols to derive more mature cell populations. A third benefit for using expanded progenitor cells is that such cultures are often quite homogenous. In contrast, hPSC cultures instructed to differentiate into a specific lineage generally contain other cell types. Therefore, the yield of more mature cell types upon subsequent differentiation is higher when starting with a homogenous, lineage restricted cell population than when starting with undifferentiated hPSCs.

The conditions that we developed for the culture and expansion of IMP cells are fully defined and free from animal-derived components, which will be important when cells are intended for therapeutic applications. Moreover, these optimized conditions are robust, as demonstrated by their ability to support derivation and expansion of IMP cells from two hES (H9 and Hues9) and one hiPS (RiPS) cell lines. Additionally, IMP cells grown in these optimized conditions can be frozen and thawed without any detectable effect on proliferative capacity or differentiation potential. Finally, these optimized conditions allow for near unlimited expansion to quantities ($\sim 10^{20}$) necessary for drug screening or regenerative medicine purposes (*Chen et al., 2013*).

Expandable lineage restricted cell populations have been developed for other lineages, including the neural and EN lineages. Several protocols have been described for the derivation of neural progenitor (NP) cells, which can proliferate extensively and differentiate into all the neural lineages and supporting cells (neurons, astrocytes, and oligodendrocytes) that compromise the central nervous system (*Reubinoff et al., 2001*; *Shin et al., 2006*; *Chambers et al., 2009*). EN progenitor (EP) cells represent another example of lineage restricted progenitor cells (*Cheng et al., 2012*). These cells retain the ability to differentiate into endodermally derived tissues, including liver and pancreas. Interestingly, differentiation into functional beta-cells is greatly improved when starting with EP cells compared to undifferentiated hPSCs.

Although both EP and IMP cells exhibit restriction with respect to their developmental potency, IMP cells are more severely restricted as they fail to produce certain mesodermally-derived cell populations, such as blood and heart muscle. We currently do not understand the mechanism by which the culture conditions defined for the derivation and expansion of IMP cells lead to this highly restricted developmental potential. During embryogenesis, as the mesoderm emerges and migrates from the primitive streak, it is further specified into PM, LM, and IM. Interestingly, both FGF and WNT/β-catenin signaling regulate this ME cell specification, migration, and proliferation (*Ciruna and Rossant, 2001*; *Sweetman et al., 2008*; *Aulehla and Pourquie, 2010*).

Along similar lines, modulation of the certain signaling pathways, such as WNT, can further refine and specify the differentiation potential of hPSC-derived progenitors. For example, we previously showed that levels of WNT/β-catenin signaling instruct the positional identity of NPCs and, upon subsequent differentiation, of the resulting neuronal cell population (*Moya et al., 2014*). Specifically, high levels of WNT signaling instructed NP cells to adopt a posterior fate, consistent with WNT's role in posterior patterning during development. In a separate study, the level of WNT activation achieved through GSK3-β inhibition was found to directly influence the ME subtype of differentiating hPSCs

(*Mendjan et al., 2014*). We speculate that continuous activation of the WNT and FGF signaling pathways is acting not only to stabilize the IMP cell state, but also to restrict its differentiation potential to cell types derived from the IM lineage.

The development of lineage-restricted progenitors offers an opportunity to investigate mechanisms by which specific developmental stages can be paused. Recent studies to profile epigenetic changes during the differentiation of hPSCs to pancreatic beta cells indicate that specific chromosomal regions open during specific windows of differentiation, thereby conferring a certain developmental competence to sequentially acquire increased lineage restriction (*Wang et al., 2015*). In the future, the intermediate mesodermally restricted cell population described here can provide a further window into the mechanisms by which developmental competence is established and maintained.

# Materials and methods

## hPSC culture

Human ES cell lines H9 and Hues9 were obtained from WiCell and Harvard University, respectively. All experiments described in this study were approved by a Stem Cell Research Oversight Committee (Protocol #100210ZX, PI Willert). The human induced pluripotent stem cell line BJ RiPS (*Warren et al., 2010*) was provided under a Material Transfer Agreement from Dr D Rossi (Childrens Hospital Boston, MA, United States). The H9 line carrying GFP in the SOX17 locus (*Wang et al., 2011*) was provided under a Material Transfer Agreement from Dr Seung Kim (Stanford School of Medicine). The following media were used: BJ RiPS and Hues 9 ES (DMEM/F12 mixed, 20% (vol/vol) Knockout Serum Replacement, 1% (vol/vol) penicillin-streptomycin, 1% (vol/vol) nonessential amino acids, 2 mM L-glutamate, 0.1 mM β-mercaptoethanol and 10 ng/ml FGF2 (PeproTech)); H9 ES (DMEM/F12 supplemented with L-Ascorbic Acid, Selenium, Transferrin, NaHCO$_3$, Insulin, TGFβ1, and FGF2 as described previously (*Chen et al., 2011*)). Fresh media was added daily to all cells. Every 5 days, colonies were enzymatically passaged with Accutase (Thermo Fisher Scientific, Waltham, MA, United States) and transferred to a Matrigel-coated culture dish. All media components are from Thermo Fisher Scientific unless indicated otherwise. For all experiments, hPSCs were used between passages 20 and 50 in this study.

## Array fabrication and characterization

ACME slides were fabricated as previously described (*Brafman et al., 2012*). Briefly, glass slides were cleaned, silanized, and then functionalized with a polyacrylamide gel layer. For ECMP arrays, stock solutions of ECMPs were suspended at 250 µg/ml in ECMP printing buffer (100 mM acetate, 5 mM EDTA, 20% [vol/vol] glycerol and 0.25% [vol/vol] Triton X-100, pH 5.0). ECMP solutions were mixed in all possible 128 combinations in a 384-well plate. For GF and SM arrays, stock solutions were suspended at 1 mg/ml in soluble factor printing buffer (100 mM acetate, 5 mM EDTA, 19% glycerol [vol/vol] and 0.25% [vol/vol] Triton X-100, 10 mM trehalose dehydrate [Sigma], 1% poly(ethylene glycol), pH 5). GF solutions were then mixed into 400 combinations representing all single, pairwise, and non-redundant three-way combinations possible in a 384-well plate. The following ECMPs, GFs, and SMs (Product/Vendor/Catalog #/Concentration) were used: Collagen I/Sigma–Aldrich (St. Louis, MO, United States)/C7774/250 µg/ml, Collagen III/Sigma–Aldrich/C4407/250 µg/ml, Collagen IV/Sigma–Aldrich/C7521/250 µg/ml, Collagen V/Sigma–Aldrich/C3657/250 µg/ml, Fibronectin/Sigma–Aldrich/F2518/250 µg/ml, Laminin/Sigma–Aldrich/L6274/250 µg/ml, Vitronectin/Sigma–Aldrich/V8379/250 µg/ml, Wnt3a/In House/100 ng/ml, R-Spondin/In House/100 ng/ml, CHIR98014/Selleck Chemicals (Houston, TX, United States)/S2745/50 ng/ml, Dkk-1/R&D Systems (Minneapolis, MN, United States)/5439-DK-010/50 ng/ml, IWP-2/Tocris (United Kingdom)/3533/50 ng/ml, FGF/Thermo Fisher Scientific/13256-029/40 ng/ml, KGF/Thermo Fisher Scientific/PHG0094/50 ng/ml, VEGF/R&D Systems/293-VE-010/50 ng/ml, EGF/R&D Systems/236-EG-01M/50 ng/ml, SHH/R&D Systems/464-SH-025/50 ng/ml, Cyclopamine/Tocris/1523/50 ng/ml, BMP4/R&D Systems/314-BP-010/50 ng/ml, Activin/R&D Systems/338-AC-010/50 ng/ml, Dorsomorphin/Sigma–Aldrich/P5499-5MG/50 ng/ml, SB 431542/Tocris/1614/50 ng/ml, Noggin/R&D Systems/6057-NG-025/50 ng/ml. The hit ECMP condition from the primary screen was used as a substrate to print the GFs and SMs in the second screen. 20 individual spots of each protein/GF/SM mixture, clustered into groups of five and printed in different quadrants of the slide, were deposited with a 450 µm pitch on the acrylamide gel pad using a

SpotBot Personal Microarray Printer (ArrayIt, Sunnyvale, CA, United States) equipped with Stealth SMP 4.0 split pins. The pins were cleaned by sonication in 5% Micro Cleaning Solution (ArrayIt) and dH$_2$O immediately before use. Between each sample in the source plate, the pins were dipped in a 50% DMSO and water solution, washed for 25 s with dH$_2$O and dried.

## Slide imaging, quantification, and analysis

Slides were fixed with 4% PFA for 10 min at room temperature (RT) and washed with PBS. Slides were imaged using the CellInsight CX5 High Content Screening (HCS) Platform (Thermo Fisher Scientific). The system was programmed to visit each spot on the array, perform autofocus, and acquire DAPI and FITC (GFP). Cell counts and stain intensities were measured using Thermo Fisher Scientific HCS Studio 2.0 Software using the built-in object identification and cell intensity algorithms.

## MP/IMP cell derivation and culture

Undifferentiated hPSCs were re-plated on Matrigel at a density of $3 \times 10^3$ cells/cm$^2$ and cultured in ES cell culture medium for 4 days. To direct cells to the mesoderm lineage, the media was switched to serum free differentiation media (consisting of RPMI 1640, 1× B27 minus Insulin, and 1% [vol/vol] penicillin-streptomycin). Cells were treated with 10 µM CHIR-98014 (CHR, Tocris) for the first 24 hr and then allowed to recover for an additional 24 hr without CHR. Tissue culture plates were incubated with ECMP coating buffer (PBS with 15 ng/ml Collagen I [C1], 15 ng/ml Collagen III [C3], 15 ng/ml Collagen IV [C4], 50 ng/ml FN, 15 ng/ml VN) overnight at 37° with volume sufficient to coat the surface area of the well. Mesoderm (48 hr) cells were single-cell passaged with Accutase and replated onto C1 C3 C4 FN VN-coated plates at a density of $3.5 \times 10^3$ cells/cm$^2$ in serum free differentiation media supplemented with 1 µM CHR and 20 ng/ml FGF. Media was also supplemented with 10 µM Y27632 (Wako, Richmond, VA, United States) for improve passaging efficiency. Optimal CHR concentration varied with cell line; Hues 9 MP cells propagated in colonies most efficiently at 0.25 µM while BJ RiPS cells did so at 0.05 µM. Manual picking of colonies in passage 1 improved MP/IMP expansion. Differentiated cells around colonies were scraped away before passaging. Half the media was changed the day after passaging and then full media changes were made every other day thereafter. For routine passaging, MP/IMP cell cultures reaching 85% confluency were dissociated using a 0.5 mM EDTA (in Ca$^{2+}$/Mg$^{2+}$-free PBS, pH 8.0) at RT for 5 min. MP cells were removed from the plate via gentle washing with the EDTA solution. Using this method, MP cells were routinely passaged every 5–8 days.

## Differentiation of hES cells to EN, EC and ME

### EN differentiation

Human ES cells were differentiated to EN as previously described (*D'Amour et al., 2005*). Initiated on days 4–6 after passage (depending on culture density), sequential, daily media changes were made for the entire differentiation protocol. After a brief wash in PBS (with Mg/Ca), cells were cultured in RPMI (without FBS), Activin A (100 ng/ml) and Wnt3a (25 ng/ml) (generated in house as described (*Willert, 2008*)) for the first day. The next day the medium was changed to RPMI with 0.2% vol/vol FBS and Activin A (100 ng/ml), and the cells were cultured for 2 additional days. Definitive EN was collected at day 3 for analysis.

### EC differentiation

Human ES cells were differentiated to EC by modifying an established neural rosette protocol (*Wilson and Stice, 2006*). 2 days before passaging hES cells, medium was changed to N2 medium (DMEM/F12 with 1× N2). 1 day before passaging, medium was changed to N2 medium supplemented with 1 µM of Dorsomorphin (DSM) (cat# 171261; Calbiochem/EMD Millipore, Billerica, MA, United States). The day of passaging, EBs were initiated by detaching cells with Accutase and gentle cell scraping. 2e6 H9 cells were used to seed one well of a 6-well low binding plate and placed on a rotating platform (95 rpm) in a 37°C incubator. 2 days later, medium was changed to N2 medium with 1 µM DSM and media changes were made as needed until 8 days after EB formation, at which point EBs were replated onto Matrigel-coated plates using NBF media (DMEM/F12 with 0.5× N2, 0.5× B-27, 20 ng/ml of FGF and 1% P/S) to form rosettes. 4–6 days after plating onto Matrigel, cells were collected for analysis.

## Mesoderm differentiation

Human ES cells were differentiation to mesoderm as previously described (*Lian et al., 2013*). Once hES cells were 50–60% confluent, medium was changed to serum free differentiation medium (RPMI supplemented with 1× [vol/vol] B27 [without insulin]) with 10 µM CHIR-98014. After 24 hr, the medium was changed to serum free differentiation medium without CHIR-98014. Cells were collected at 48 hr for analysis.

# Human ES and IMP cell differentiation

## Hematopoietic differentiation

Human ES and IMP cells were differentiated to the hematopoietic lineage as previously described (*Ng et al., 2008*). ES cells were differentiated towards hematopoietic precursors first by ME induction with 25 ng/ml human BMP4 for 4 days. IMP cells were treated as d4 ME, bypassing BMP4 treatment. After mesoderm induction, cells were treated with 20 ng/ml FGF and 50 ng/ml human VEGF (Humanzyme) for 4 days and then with 50 ng/ml Flt-3L (R&D Systems) and 150 ng/ml IL-6 (R&D Systems) for 4 days.

## CM differentiation

Human ES and IMP cells were differentiated to the cardiac lineage as previously described (*Lian et al., 2012*). Human ES cells were induced to mesoderm with 10 µM CHIR 98014 for 24 hr, then incubated for 48 hr in serum free differentiation media. IMP cells were treated as d3 cultures, bypassing this initial treatment. Cells were then treated with IWP-2 for 4 days, incubated for an additional 2 days in serum free differentiation media, then supplemented with insulin at day 9.

## Kidney differentiation

Human ES and IMP cells were differentiation to the kidney lineage as previously described (*Taguchi et al., 2014*). In serum free differentiation media (SFDM; DMEM/F12 supplemented with 2% (vol/vol) B27 (without retinoic acid), 2 mM L-glutamine, 1% (vol/vol) ITS, 1% (vol/vol) nonessential amino acids, 90 µM β-mercaptoethanol and 1% (vol/vol) penicillin/streptomycin), hES and MP cells were aggregated at 10,000 cells per well in U-bottom 96-well low-cell-binding plates to form EBs. EBs were formed in the presence of 10 µM Y27632 (Wako) and 0.5 ng/ml human BMP4 (Stemgent, Lexington, MA, United States). After 24 hr, the SFDM was supplemented with 1 ng/ml human Activin A and 20 ng/ml human FGF2. After 48 hr, the SFDM was supplemented with 0.5 ng/ml BMP4 and 10 µM CHIR. IMP cells were treated as d3 cultures, bypassing this initial treatment. Subsequently, half of the culture medium volume was refreshed with new SFDM every other day. On day 9, the medium was changed to SFDM supplemented with 1 ng/ml human Activin A, 0.5 ng/ml BMP4, 3 µM CHIR, and 0.1 µM retinoic acid. On day 11, the medium was changed to SFDM containing 1 µM CHIR and 5 ng/ml FGF9. All data shown are representative examples of at least three independent experiments.

# Quantitative RT-PCR

RNA was isolated using RNeasy Plus Micro Kit (Qiagen, Germany) reverse-transcribed with random primers and qScript cDNA Supermix (Quanta, Gaithersburg, MD, United States). Before reverse transcription, 5 µg of RNA was digested by RNase-free DNase I (Ambion/Thermo Fisher Scientific) to remove genomic DNA. qPCR was carried out using a Real-Time PCR System (Bio-Rad, Hercules, CA, United States) and Taqman qPCR Mix with a 10-min gradient to 95℃ followed by 40 cycles at 95℃ for 15 s and 60℃ for 1 min. The following Taqman (Thermo Fisher Scientific) gene expression assay primers (Gene/ABI Assay #) were used: 18s/Hs99999901_s1, OCT4/Hs04260367_gH, NANOG/Hs04399610_g1, SOX2/Hs01053049_s1, FOXA2/Hs00232764_m1, SOX1/Hs01057642_s1, MESP1/Hs01001283_g1, MIXL1/Hs00430824_g1, LHX1/Hs00232144_m1, PDGFRA/Hs00998018_m1, PAX1/Hs01071293_g1, TBX6/Hs00365539_m1, TCF15/Hs00231821_m1, MEOX1/Hs00244943_m1, NKX2.5/Hs00231763_m1, ISL1/Hs00158126_m1, LMO2/Hs00153473_m1, KDR/Hs00911700_m1, PAX2/Hs01057416_m1, EYA1/Hs00166804_m1, SALL1/Hs01548765_m1, OSR1/Hs01586544_m1, LHX1/Hs00232144_m1, WT1/Hs01103751_m1, CITED2/Hs01897804_s1, PECAM1/Hs00169777_m1, HOXC9/Hs00396786_m1, ITGA8/Hs00233321_m1, PBX1/Hs00231228_m1, HOXA10/Hs00172012_m1, HOXA11/Hs00194149_m1, GDNF/Hs01931883_s1, FOXD1/Hs00270117_s1, SIX2/Hs00232731_m1, CDX2/Hs01078080_m1, FGF5/Hs03676587_s1. Gene expression was normalized to 18S rRNA levels. Delta $C_t$ values were calculated as $C_t^{target} - C_t^{18s}$. All experiments were performed with three technical replicates. Relative fold changes in gene expression were calculated using the $2^{-\Delta\Delta C_t}$ method (*VanGuilder et al., 2008*).

## Antibodies

The following antibodies were used (Antibody/Vendor/Catalog #/Concentration): Rabbit anti-NANOG/Santa Cruz Biotechnology (Dallas, TX, United States)/SC-33759/1:50, Rabbit anti-OCT4/Santa Cruz/SC-9081/1:50, Mouse anti-MIXL1/R&D Systems/MAB2610/1:200, Mouse anti-PAX2/Creative Diagnostics (Shirley, NY, United States)/DMABT-H14539/1:200, Rabbit anti-SIX2/Abcam (Cambridge, MA, United States)/ab68908/1:200, Rabbit anti-WT1/Santa Cruz Biotechnology/sc-192/1:200, Rabbit anti-SALL1/Abcam/ab31526/1:200, Mouse anti-E Cadherin/Abcam/ab1416/1:200, Mouse anti-Human Nuclear Antigen/Abcam/ab191181/1:250, Goat anti-FOXD1/Santa Cruz Biotechnology/sc-47585/1:200, Rabbit anti-Ki67/Abcam/ab15580/1:250, APC anti-human CD56 (NCAM)/BioLegend (San Diego, CA, United States)/318309/5 µl per test, PE anti-human CD326 (EpCAM)/BioLegend/324205/5 µl per test, Alexa-647 Mouse IgG2a Isotype Control/BD/558053/20 µl per test, PE Mouse IgG1 Isotype Control/BioLegend/400113/5 µl per test, PE Mouse IgG2a Isotype Control/BD Biosciences (San Jose, CA, United States)/561552/5 µl per test, Alexa 647 Donkey Anti-Goat/Thermo Fisher Scientific/A-21447/1:200, Alexa 647 Donkey Anti-Rabbit/Thermo Fisher Scientific/A-31573/1:200, Alexa 647 Donkey Anti-Mouse/Thermo Fisher Scientific/A-31571/1:200, Alexa 546 Donkey Anti-Rabbit/Thermo Fisher Scientific/A-10040/1:200, Alexa 546 Donkey Anti-Mouse/Thermo Fisher Scientific/A-10036/1:200, Alexa 488 Streptavidin Conjugate/Thermo Fisher Scientific/S-11223/1:200, Alexa 488 Donkey Anti-Rabbit/Thermo Fisher Scientific/A-21206/1:200, Alexa 488 Donkey Anti-Mouse/Thermo Fisher Scientific/A-21202/1:200.

## Flow cytometry

Cells were dissociated with Accutase (Thermo Fisher Scientific) at 37°C for 4 min and triturated using fine-tipped pipettes. For intracellular antibody staining, cells were fixed for 15 min with Cytofix (BD Biosciences), washed twice with flow cytometry buffer (PBS, 1 mM EDTA, and 0.5% FBS), permeabilized with Cytoperm (BD Biosiences) for 30 min on ice, and washed twice with flow cytometry buffer, and resuspended at a maximum concentration of $5 \times 10^6$ cells per 100 µl. Cells were incubated with primary antibodies on ice for 1 hr, washed twice with flow cytometry buffer. If necessary, cells were incubated with secondary antibodies on ice for 1 hr and then washed three times. After passing through a 40 µm cell strainer, cells were resuspended in flow cytometry buffer at a final density of $2 \times 10^6$ cells ml$^{-1}$. Propidium iodide (Sigma) was added at a final concentration of 50 mg ml$^{-1}$ to exclude dead cells. Cells were analyzed on the FACS Fortessa (Becton Dickinson). For each sample, at least three independent experiments were performed. Results were analyzed using FlowJo software.

## Immunocytochemistry

Monolayer cultures were gently washed with PBS prior to fixation. Cultures were fixed for 10 min at 4°C with fresh paraformaldehyde (4% [wt/vol] in PBS). For sectioning aggregates of cells in suspension, samples were fixed with 4% paraformaldehyde, embedded in optimal cutting temperature compound (Tissue Tek) and cryo-sectioned at 10-µm thickness before staining, Cells were blocked and permeabilized with 2% (wt/vol) BSA, 0.2% ([vol/vol] in PBS) Triton X for 30 min at RT. Cells were then washed twice with PBS. Primary antibodies were incubated overnight at 4°C and washed twice with PBS. Secondary antibodies were incubated for 1 hr at 37°C. Antibodies used are as listed above. Prior to imaging, samples were stained with DAPI for 10 min, washed and mounted in Vectashield (Vector Laboratories, Burlingame, CA, United States), covered with coverslips, and sealed with nail polish. Images were taken using an Olympus FluoView1000 multi-photon confocal microscope. All IF analyses were repeated a minimum of three times and representative images are shown.

## High throughput RNA-seq

Total RNA was isolated from cells, depleted of genomic DNA and rRNA and fragmented to ∼200 bp by RNase III. After ligating the Adaptor Mix, fragmented RNA was converted to the first strand cDNA by ArrayScript Reverse Transcriptase (Ambion/Thermo Fisher Scientific), size selected (100–200 bp) by gel electrophoresis, and amplified by PCR using adaptor-specific primers. Deep sequencing was performed on an Illumina (San Diego, CA, United States) Genome Analyzer II. Analysis of genome-wide expression data was performed as previously described (Trapnell et al., 2012, 2013). Briefly, raw reads were aligned to the reference human genome (hg19) using TopHat. Cufflinks was used to assemble individual transcripts from the mapped reads. Cuffmerge was used to merge the assembled transcripts from the two biologically independent samples. Cuffdiff was used to calculate gene

expression levels and test for the statistical significance of differences in gene expression. Reads per kilobase per million mapped reads were calculated for each gene and used as an estimate of expression levels. The full RNA-seq data set for the IMP cells is provided in *Supplementary file 2*.

## SC co-culture assay

hES or IMP-derived MM cells were cultured with mouse embryonic SC taken from E11.5 or E12.5 embryos at the air-fluid interface on a polycarbonate filter (0.8 mm; Whatman/Sigma-Aldrich) fed with DMEM containing 10% fetal calf serum, as described previously (*Kispert et al., 1998*; *Osafune et al., 2006*; *Gallegos et al., 2012*; *Martovetsky et al., 2013*).

## Re-aggregation assay

The re-aggregation assay was performed as previously described (*Unbekandt and Davies, 2010*; *Davies and Chang, 2014*). To prepare the kidney tissue for recombination, embryonic kidneys from 12.5–13.5-dpc (days post coitum) mice were isolated and dissected free of surrounding tissues as previously described (*Gallegos et al., 2012*; *Martovetsky et al., 2013*). Briefly, embryonic kidneys were digested with trypsin at 37°C for 10 min and dissociated by manually pipetting. After the cells had been filtered through a 100 μm cell strainer, $4$–$10 \times 10^5$ embryonic kidney cells were recombined with 4% (by number) of hESC- or IMP-derived cells and then centrifuged at $400 \times g$ for 2 min to form a pellet. The pellet was allowed to aggregate by culturing in DMEM supplemented with 10% FBS overnight in a sterile PCR tube. The following day, the aggregate was transferred to the top of a Transwell polycarbonate filter (0.4 μm pore size). The filter was placed with the well of a 12-well dish to which DMEM supplemented with 10% FBS was added to bottom of the well. The aggregate was then cultured for 4 days at the air-fluid interface before fixation and analysis. The following lectins were used to stain organoid cultures (Lectin/Vendor/Catalog #/Concentration): Biotinylated DBA/Vector Laboratories/B-1035/1:200, LTL, Biotinylated-LTL/Vector Labs/B-1325/1: 200. An Alexa 647 Streptavidin Conjugate (Thermo Fisher Scientific; S-21374) was used at 1:200 for detection of these lectins.

## Teratoma/Transplantation assay

For the subcutaneous injection, H9 hES or MP cells were dissociated, mixed with 250 μl Matrigel, and transplanted subcutaneously into the thigh and shoulder of nude mice. Each mouse received two injections of cells, one near the front legs and one near the hind legs. Teratoma formation was monitored over a period of 4–12 weeks. A total of 3 mice were injected with $0.5 \times 10^6$ hES cells per site. All six injection sites yielded teratomas of 10 mm or greater. Another 6 mice were injected with MP cells: 2 mice received $0.5 \times 10^6$ cells per injection, 2 mice received $0.75 \times 10^6$ cells per injection, and 2 mice received $1.0 \times 10^6$ cells per injection. Of the 12 injection sites, only one site maintained a small lump of 1 mm that did not grow in size. No MP cell injection yielded a growth of the size observed for hES cells. All animal work was approved by the institutional IACUC committee (Protocol Number S06321, PI Willert).

## Chromosome counting

Karyotype analysis was performed by Cell Line Genetics, Inc., Madison, Wisconsin, United States. For each submitted MP cell line (H9 at passage 10 and H9_SOX17-GFP at passage 15 [*Wang et al., 2011*], kindly provided by Dr Seung Kim, Stanford School of Medicine, Palo Alto, CA, United States chromosome numbers were determined for 20 cells using G-banded metaphase spreads.

## Statistical analyses

All averaged data are expressed ±standard error of the mean of three independent biological replicates unless otherwise stated. For comparisons of discrete data sets, unpaired Student's *t*-tests were performed to calculate p-values between experimental conditions and controls and a p-value <0.05 was considered statistically significant. For each ACME experiment, the ratio ($R_i$) of the $\log_2$ of the T-GFP signal and the DNA signal was calculated for each spot. From this a differentiation *z*-score was calculated for each spot $Z_{DIF} = (R_i - \mu_{DIF})/\sigma_{DIF}$, where $R_i$ was the ratio for the spot, $\mu_{DIF}$ was the average of the ratios for all spots on each array, and $\sigma_{DIF}$ was the S.D. of the ratios for all spots on each array. Differentiation *z*-scores from replicate spots ($n = 5$ per condition) were averaged for each ECMP condition on the array. The replicate average *z*-scores were displayed in a heat map with rows corresponding to individual conditions and columns representing independent array experiments ($n = 5$ for each replicate). For each array experiment, all columns were mean-centered and normalized to one unit S.D. The rows were clustered

using Pearson correlations as a metric of similarity. All clustering was performed using Gene Cluster. The results were displayed using a color code with red and green representing an increase and decrease, respectively, relative to the global mean. All heat maps were created using Tree View. Global main effects principal component analysis was performed as previously described (*Box et al., 2005*).

## Acknowledgements

We are grateful to Dr M Mercola (UCSD and Sanford Consortium for Regenerative Medicine) for kindly providing H9 cells carrying the T-GFP reporter cassette, to Drs Juan Carlos Belmonte and Zhongwei Li (Salk Institute for Biological Studies) for providing human fetal kidney RNA. This work was supported by grants from the California Institute for Regenerative Medicine (RB3-05086), the National Institute of Health (U01-DK089567), the UCSD Stem Cell Program, a gift from Michael and Nancy Kaehr, and was made possible in part by the CIRM Major Facilities grant (FA1-00607) to the Sanford Consortium for Regenerative Medicine.

## Additional information

### Funding

| Funder | Grant reference | Author |
| --- | --- | --- |
| California Institute for Regenerative Medicine (CIRM) | RB3-05086 | Nathan Kumar, Jenna Richter, David Brafman, Karl Willert |
| National Institute of Diabetes and Digestive and Kidney Diseases (NIDDK) | U01-DK089567 | Nathan Kumar, David Brafman, Karl Willert |

The funders had no role in study design, data collection and interpretation, or the decision to submit the work for publication.

### Author contributions

NK, KW, Conception and design, Acquisition of data, Analysis and interpretation of data, Drafting or revising the article; JR, JC, CT, Acquisition of data, Analysis and interpretation of data; KTB, Conception and design, Acquisition of data, Analysis and interpretation of data; SKN, DB, Conception and design, Analysis and interpretation of data, Drafting or revising the article; TG, Conception and design, Acquisition of data, Analysis and interpretation of data, Contributed unpublished essential data or reagents

### Author ORCIDs

Kevin T Bush, http://orcid.org/0000-0001-5889-2847

### Ethics

Animal experimentation: This study was performed in strict accordance with the recommendations in the Guide for the Care and Use of Laboratory Animals of the National Institutes of Health. All of the animals were handled according to approved institutional animal care and use committee (IACUC) protocols of the University of California San Diego. The protocol was approved by the Committee on the Ethics of Animal Experiments of the University of Minnesota (Protocol Number S06321). Every effort was made to minimize suffering.

## Additional files

### Supplementary files

• Supplementary file 1. This table provides the complete list of genes with similar expression levels in MP cells and mesoderm (MP + ME, panel A), and of genes expressed only in MP cells (MP only, panel B).

• Supplementary file 2. This file provides the complete RNA-seq data set of MP cells at passage 10. Column A provides gene names. Values in column B are RPKM.

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
