## [Decision Letter]

Thank you for submitting your work entitled “Generation of an expandable intermediate mesoderm restricted progenitor cell line from human pluripotent stem cells” for peer review at *eLife*. Your submission has been favorably evaluated by Janet Rossant (Senior Editor) and three reviewers, one of whom, Amy Wagers, is a member of our Board of Reviewing Editors.

The reviewers have discussed the reviews with one another and the Reviewing editor has drafted this decision to help you prepare a revised submission.

The authors report on a new system to derive mesodermal progenitor cells from human pluripotent stem cells. They use an arrayed cellular microenvironments (ACME) system to identify and optimize culture conditions to support derivation of this population of cells, and show that this MP population differentiates readily along the renal lineage, but does not generate blood or cardiac cell fates. Overall, the reviewers found the manuscript interesting and the data of high quality. The reviewers agreed that the cellular microarray technology employed is clever and of potential broad interest to other groups seeking to generate defined cell types from pluripotent cells, as establishing appropriate in vitro conditions to “capture” the desired cell fate is often a bottleneck for such endeavors. While the ACME system itself has been previously described (3), its application to hPSC differentiation has not previously been reported. Also, the new protocol described for the generation of expandable mesodermal precursors is novel, and likely to be useful for a number of other laboratories and studies. However, all three reviewers raised concerns that the testing done thus far of renal differentiation is inadequate. A more stringent assessment of the potential of these cells is needed for publication in *eLife*, and the reviewers have provided specific suggestions to address this point below.

Essential revisions:

1) The experiments testing differentiation capacity to renal lineage are not entirely convincing. Using a single cell disassociation and re-aggregation assay, the authors show that the MP cells incorporate around a cap-mesenchyme-like structure (Figure 8). The problem is that this by no means shows renal lineage differentiation. The MP have a location consistent with FoxD1^+^ mesenchyme, which go on to develop into renal stroma. But there is no co-staining for FoxD1 to prove that these cells have actually differentiated into the renal lineage. Nor is there co-staining for Six2 to delineate the location of the cap mesenchyme. It is surprising that the MPs apparently do not integrate into the Six2^+^ cap mesenchyme – which should be present in this reaggregation assay – given that the authors show the potential of these cells to express Six2, or at least a fraction of them. The embryonic rat spinal cord assay results are similarly hard to interpret. ECAD and LTL expression are evident, but there is no clear tubule-like structure. Others have published very nephron-like structures using this assay, and even glomerulus-like structures. The Six2 expression is punctate, and scattered across the cell population. In nephrogenesis, Six2^+^ cells aggregate adjacent to one another, in the cap mesenchyme sitting on top of the ureteric-bud. These structures have all been published by others using the two assays shown here, but there is no real incorporation of the MP cells into nephron-like structures. The study therefore requires better evidence of true differentiation into renal lineage and incorporation of MP-derived cells into nephron-like structures in order to prove that this new line has actual renal potential and should examine the histogenetic potential of the MP-derived cells in both types of explant systems employed. A more informative study would be to harvest the MP-derived cells from the two types of explants and ascertain the differentiation endpoints of these cells by transcriptome analysis (e.g. RNA-Seq) against the kidney cells (as for the analysis of the MM cells in vitro, Figure 7). Additionally, the developmental potency of the MP cells may be studied by testing if MP cells alone can generate both metanephric mesenchyme and ureteric bud/nephron progenitors, as shown previously for hESCs in other studies. The authors should consider the type of analysis published by Little (Nat Cell Bio 2014 January;16(1):118–26) and Belmonte (Nat Cell Bio 2013 December;15(12):1507–15). In these studies, researchers took their candidate cells and, using the same reaggregation assays these authors used in Figure 8, clearly showed by high power confocal, that their human ES-derived cells integrated into differentiated structures. In Little's case, into Six2^+^ cap mesenchyme as well as into a tubule-like structure. At present, the current manuscript shows no integration – just the presence of HuNu^+^ cells around an epithelial structure. This reaggregation assay needs to be pushed further, i.e., (1) to generate real tubules and determine if any cells integrate, (2) to co-stain for cap mesenchyme (Six2) and determine if the cells integrate there, and (3) to co-stain for stromal progenitors (FoxD1) and ask if cells integrate there. Alternative, but also less stringent options, might be to extend the spinal cord assay in Figure 8 to determine if tubules are formed that contain lumens, polarized expression. These studies must use high power images, as the low power ones available in the current manuscript are insufficient to discriminate the morphological features. Finally, if it proves technically difficult to evaluate the morphogenetic potential of the MP-derived cells in these explants, an in-depth molecular profiling of the MP-derived cells in the explants could be performed to establish the identity of these cells at the endpoint of the culture experiment.

2) Regarding differentiation of MP cells along non-renal lineages (Figure 6), it would be useful to extend the analysis past day 14 to test the possibility that MP cells are simply delayed in their differentiation along these lineages.

[Editors' note: further revisions were requested prior to acceptance, as described below.]

Thank you for resubmitting your work entitled “Generation of an expandable intermediate mesoderm restricted progenitor cell line from human pluripotent stem cells” for further consideration at *eLife*. Your revised article has been favorably evaluated by Janet Rossant (Senior Editor), Amy Wagers (Reviewing Editor), and two reviewers. The manuscript has been improved but there are some remaining issues that need to be addressed before acceptance, as outlined below:

The reviewers appreciate the inclusion of additional data related to the differentiation potential of the derived IMP cells. These results clearly define the limited potential of the IMP cells as characterized thus far, which appear to express various markers characteristic of various stages of differentiation along a nephrogenic pathway, but are incapable of forming tubules or nephron-like structures. Still, the reviewers recognize the merit of the manuscript's description of these cells, particularly as they are human, and the potential broad utility of the matrix conditions identified, which are likely to inform other groups' pluripotent cell differentiation approaches.

However, in order to be acceptable for publication in *eLife*, the authors must provide more convincing data demonstrating that the derived cells express FoxD1. The data currently included in Figure 9 do not provide sufficient confidence that possible background/artifactual staining has been properly excluded. Please provide appropriate controls demonstrating the specificity of this staining and/or provide other data demonstrating unequivocally that these cells express FoxD1.

[Editors' note: further revisions were requested prior to acceptance, as described below.]

Thank you for resubmitting your work entitled “Generation of an expandable intermediate mesoderm restricted progenitor cell line from human pluripotent stem cells” for further consideration at *eLife*. Your revised article has been favorably evaluated by Janet Rossant (Senior Editor) and a Reviewing editor.

The manuscript has been improved but there are some remaining issues that need to be addressed before acceptance, as outlined below.

In particular, while the arrowheads are helpful and support from the RT-PCR analysis is noted, the data still are not entirely convincing. It is essential to provide appropriate control stains (e.g., isotype controls) demonstrating absence of background for FoxD1 and HuNu data presented in Figure 9. These can be included in the supplement.

---

## [Author Response]

Essential revisions:

*1) The experiments testing differentiation capacity to renal lineage are not entirely convincing. Using a single cell disassociation and re-aggregation assay, the authors show that the MP cells incorporate around a cap-mesenchyme-like structure (*Figure 8*). The problem is that this by no means shows renal lineage differentiation. The MP have a location consistent with FoxD1*^*+*^
*mesenchyme, which go on to develop into renal stroma. But there is no co-staining for FoxD1 to prove that these cells have actually differentiated into the renal lineage. Nor is there co-staining for Six2 to delineate the location of the* cap *mesenchyme. It is surprising that the MPs apparently do not integrate into the Six2*^*+*^ cap *mesenchyme – which should be present in this reaggregation assay – given that the authors show the potential of these cells to express Six2, or at least a fraction of them. The embryonic rat spinal cord assay results are similarly hard to interpret. ECAD and LTL expression are evident, but there is no clear tubule-like structure. Others have published very nephron-like structures using this assay, and even glomerulus-like structures. The six2 expression is punctate, and scattered across the cell population. In nephrogenesis, Six2*^*+*^
*cells aggregate adjacent to one another, in the* cap *mesenchyme sitting on top of the ureteric-bud. These structures have all been published by others using the two assays shown here, but there is no real incorporation of the MP cells into nephron-like structures. The study therefore requires better evidence of true differentiation into renal lineage and incorporation of MP-derived cells into nephron-like structures in order to prove that this new line has actual renal potential and should examine the histogenetic potential of the MP-derived cells in both types of explant systems employed. A more informative study would be to harvest the MP-derived cells from the two types of explants and ascertain the differentiation endpoints of these cells by transcriptome analysis (e.g. RNA-Seq) against the kidney cells (as for the analysis of the MM cells* in vitro*,*
Figure 7*). Additionally, the developmental potency of the MP cells may be studied by testing if MP cells alone* can *generate both metanephric mesenchyme and ureteric bud/nephron progenitors, as shown previously for hESCs in other studies. The authors should consider the type of analysis published by Little (Nat Cell Bio 2014 January;16(1):118–26) and Belmonte (Nat Cell Bio 2013 December;15(12):1507–15). In these studies, researchers took their candidate cells and, using the same reaggregation assays these authors used in*
Figure 8*, clearly showed by high power confocal, that their human ES-derived cells integrated into differentiated structures. In Little's case, into Six2*^*+*^ cap *mesenchyme as well as into a tubule-like structure. At present, the current manuscript shows no integration – just the presence of HuNu*^*+*^
*cells around an epithelial structure. This reaggregation assay needs to be pushed further, i.e., (1) to generate real tubules and determine if any cells integrate, (2) to co-stain for* cap *mesenchyme (Six2) and determine if the cells integrate there, and (3) to co-stain for stromal progenitors (FoxD1) and ask if cells integrate there. Alternative, but also less stringent options, might be to extend the spinal cord assay in*
Figure 8
*to determine if tubules are formed that contain lumens, polarized expression. These studies must use high power images, as the low power ones available in the current manuscript are insufficient to discriminate the morphological features. Finally, if it proves technically difficult to evaluate the morphogenetic potential of the MP-derived cells in these explants, an in-depth molecular profiling of the MP-derived cells in the explants could be performed to establish the identity of these cells at the endpoint of the culture experiment.*

Thank you for these helpful comments on how to improve the experiments showing that MP (renamed to IMP) cells are capable of efficient renal differentiation. We have included additional immunofluorescence (IF) data that clearly demonstrate that metanephric mesenchyme (MM) derived from IMP cells efficiently incorporate into the mesenchyme surrounding tubule-like structures. In addition, we show that these cells co-stain for FOXD1 and the lectin LTL, both markers of kidney mesenchyme. We made the following changes to the manuscript to include this new data:

A) Figure 8 now shows the spinal cord induction assay. This experiment is informative as it shows that the IMP cells differentiate to express mature markers of the developing kidney, including ECAD, LTL, SALL1 and SIX2. In contrast, undifferentiated hES cells fail to express any of these markers.

B) Since the spinal cord co-culture experiments shown in Figure 8 did not produce clear nephron-like structures, we also performed kidney re-aggregation assays. This data was originally shown in Figure 8. We now show this data in Figure 9 along with the new IF results. The overall conclusion from the reaggregation assay is that IMP-derived MM cells efficiently incorporate into the mesenchyme surrounding tubule-like structures. These cells express FOXD1 and LTL. We do not see evidence of these cells incorporating into tubules or into the cap mesenchyme. As a negative control, we show that hES cells fail to incorporate into these kidney aggregates.

Taken together, we show that IMP cells can efficiently differentiate towards the kidney lineage and incorporate into the developing kidney. It can be argued that additional experiments to further characterize the renal potential of IMP cells are warranted. However, we contend that these extensive and comprehensive studies to generate and characterize these IMP cells are of significant value and interest to the scientific community and are worthy of publication at this point. Certainly, additional experiments are needed to further characterize the full differentiation potential of these IMP cells, but such experiments will be the subject of future studies.

*2) Regarding differentiation of MP cells along non-renal lineages (*Figure 6*), it would be useful to extend the analysis past day 14 to test the possibility that MP cells are simply delayed in their differentiation along these lineages.*

We have extended these differentiations beyond day 14 and have not seen any differentiation towards cardiomyocyte or hematopoietic lineages. In addition, since it is well established that differentiation into cardiomyocytes requires inactivation of Wnt signaling at a stage shortly after mesoderm induction, we have treated the IMP cells for several days with the Wnt inhibitor IWP in the absence of the GSK3 inhibitor CHIR98014 (= CHR, a Wnt agonist required for expansion and maintenance of MP cells, as illustrated in Figure 4) prior to initiating differentiation towards the cardiomyocyte lineage. These manipulations also failed to promote cardiomyocyte differentiation, further underscoring our observation that IMP cells have a highly restricted lineage potential that does not include heart or blood differentiation. We have altered the text describing Figure 6 to further emphasize these findings.

[Editors' note: further revisions were requested prior to acceptance, as described below.]

The reviewers appreciate the inclusion of additional data related to the differentiation potential of the derived IMP cells. These results clearly define the limited potential of the IMP cells as characterized thus far, which appear to express various markers characteristic of various stages of differentiation along a nephrogenic pathway, but are incapable of forming tubules or nephron-like structures. Still, the reviewers recognize the merit of the manuscript's description of these cells, particularly as they are human, and the potential broad utility of the tmatrix conditions identified, which are likely to inform other groups' pluripotent cell differentiation approaches.

*However, in order to be acceptable for publication in* eLife*, the authors must provide more convincing data demonstrating that the derived cells express FoxD1. The data currently included in*
Figure 9
*do not provide sufficient confidence that possible background/ artifactual staining has been properly excluded. Please provide appropriate controls demonstrating the specificity of this staining and/or provide other data demonstrating unequivocally that these cells express FoxD1.*

We provide additional images that clearly demonstrate that the IMP-derived cells express FOXD1. To clarify the staining of HuNu and FOXD1, we have identified individual cells with arrow heads in Figure 10. The new panel C of Figure 9 does not include these arrow heads.

In addition, in support that the IMP cells are capable of expressing FOXD1, we would also like to draw attention to Figure 7, which shows that FOXD1 expression is detectable by qPCR in IMP cells differentiated towards metanephric mesenchyme for 14 days. As expected, levels of FOXD1 expression in Day 14 MM is lower than in embryonic kidney. This data serves as independent confirmation that IMP cells are capable of generating FOXD1-positive cells.

Author response image 1.**DOI:**
http://dx.doi.org/10.7554/eLife.08413.030

[Editors' note: further revisions were requested prior to acceptance, as described below.]

The manuscript has been improved but there are some remaining issues that need to be addressed before acceptance, as outlined below.

*In particular, while the arrowheads are helpful and support from the RT-PCR analysis is noted, the data still are not entirely convincing. It is essential to provide appropriate control stains (e.g., isotype controls) demonstrating absence of background for FoxD1 and HuNu data presented in*
Figure 9*. These* can *be included in the supplement.*

We provide additional images that clearly demonstrate that the staining of FOXD1 and HuNu shown in Figure 9 is specific and real. Specifically, we show that staining of these samples with secondary antibody produces no staining, demonstrating that the FOXD1 and HuNu staining shown in Figure 9 is the result of the primary antibody, not due to non-specific secondary antibody binding. We have included these images as Figure 9—figure supplement 2 and included the appropriate in-text reference, as well as figure legend, in the amended manuscript. We would also like to draw the reviewers’ attention to the following as further demonstration of specificity of the FOXD1 and HuNu staining: (i) In Figure 9 (co-culture assays performed with IMP-derived MM) not all cells stain positive for FOXD1 and HuNu (top and middle row) and (ii) In Figure 9 (the co-culture assays performed with hES cells) none of the cells stain positive for FOXD1 and HuNu (bottom row). In conjunction with the secondary only control stains, this data shows that all positive staining for FOXD1 and HuNu is real and specific.